# NON-NEGATIVE PROBABILISTIC FACTORIZATION WITH SOURCE VARIATION

## ABSTRACT

Non-negative Matrix Factorization (NMF) is a powerful data-analysis tool to extract non-negative latent components from linearly mixed samples. It is particularly useful when the observed signal aggregates contributions from multiple sources. However, NMF only accounts for two types of variations between samples - disparities in the proportions of sources contribution and observation noise. Here, we present VarNMF, a probabilistic extension of NMF that introduces another type of variation between samples: a variation in the actual value a source contributes to the samples. We show that by modeling sources as distributions and applying an Expectation Maximization procedure, we can learn this type of variation directly from mixed samples without observing any source directly. We apply VarNMF to a dataset of genomic measurements from liquid biopsies and demonstrate its ability to extract cancer-associated source distributions that reflect inter-cancer variability directly from mixed samples and without prior knowledge. The proposed model provides a framework for learning source distributions from additive mixed samples without direct observations.

## 1 INTRODUCTION

The last few decades have brought great advances in DNA sequencing technologies, facilitating the collection of rich and diverse genomic datasets (Marguerat & Bähler (2010), Kimura (2013)). In many applications the sequenced sample represents an aggregate of multiple sources. For example, a liver tissue sample contains liver cells but also blood vessels and multiple types of immune cells. Similarly, a cancer biopsy contains tumor cells, but also a variety of other cell-types from the surrounding tissue (Haider et al., 2020). Therefore, the signals obtained from such samples represent mixed information from the multitude of cell-types that are present in the physical sample. In many applications we aim to separate the sources contributing to the mixed signal to gain insights on the underlying processes. For example, biopsies from two cancer patients may differ in the proportion of tumor cells. However, certain genes can also exhibit variation in signal within the tumor cells of the two patients (Rudin et al., 2019), constituting an alteration in the tumor-contributing signal itself. Ideally, we want to distinguish between the two types of deviations.

The challenge of decomposing mixed signals was approached from many directions (see review in Shen-Orr & Gaujoux (2013)). Most previous works rely on reference data — a molecular characterization of potential contributing sources — which is used for estimating and quantifying the proportion of each source in the mixed sample. However, by relying only on previously characterized sources, these methods are unable to identify new or inaccessible sources of signal. Additionally, these characterizations are usually determined by direct observations from the isolated sources. In many cases obtaining such direct observations is infeasible, and in others the isolation process incurs technical biases and thus estimated characterizations are non-transferable.

A different approach to decomposition is a data-driven approach of matrix factorization, employing algorithms such as Principal Component Analysis (PCA) (Jolliffe & Cadima, 2016) and Independent Component Analysis (ICA) (Comon, 1994). However, since sequencing signal is based on counts of molecules, both the mixed signal and the sources contributing to it are non-negative. Therefore, a natural model is the *Non-negative Matrix Factorization* (NMF) model (Lee & Seung, 2000), which decomposes the non-negative mixed signal into two low-rank non-negative matrices:

$$V \approx W \cdot H \qquad (1)$$

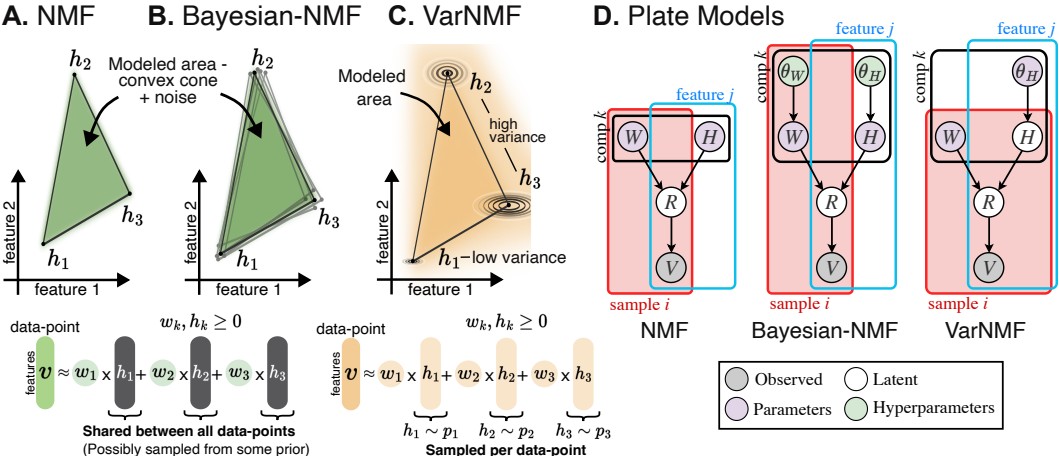

Figure 1: A-C) Illustrative examples of mixed samples with $K = 3$ non-negative components (in black) and two representative features. A) The NMF model assumptions allow us to model a convex cone between the constant components $h_1, h_2, h_3$, plus some noise. B) In Bayesian-NMF there is a prior over the location of $h_1, h_2, h_3$, C) In contrast, adding variation to the sources, each data-point has its own instantiation of sources contribution, sampled from the corresponding component distributions. This results in a wider modeled area. Separating the sources from mixed samples using NMF in this scenario will result in a solution that is wider than the original sources that created the data. D) Graphical plate models for the three models. Latent variables are ones that are integrated over in the model. Parameters are point estimate either by MLE or MAP.

where $H_k \in \mathbb{R}_{\geq 0}^M$ represents the $k$-th source as a constant component, and each sample $i = 1, ..., N$ is the result of linearly mixing these components with weights $W[i] \in \mathbb{R}_{\geq 0}^K$ plus a "noise" term. NMF serves as a powerful tool for capturing underlying structure in mixed samples. Beyond sequencing data, it is also relevant to multiple other problems such as feature extraction in computer vision (Lee & Seung, 1999), audio separation (Smaragdis & Brown, 2003), and topic modeling in NLP (Arora et al., 2012).

In the NMF model, variation between samples is due to (1) the mixing proportions $W$ and (2) observation noise. However, in many cases there are differences that are not explained entirely by these two factors. In our example above, the molecular signal of tumor cells is mostly similar, but can vary between patients, as well as in different time-points for the same patient (e.g. after receiving treatment). We can view this as tumor cells not having constant characterization, but rather as having variation in some genes signal between subjects and across time (Fig. 1A,B). Such differences in the state of the tumor cells cannot be captured or reasoned about in NMF and its Bayesian extensions.

To account for variation in sources signal between samples, we introduce a probabilistic graphical model we call VarNMF, where each source $k$ is modeled as a distribution $p_k$ instead of a constant component vector, and the parameters of these distributions and the mixing weights are estimated from data using an EM procedure (Dempster et al., 1977). This modeling also allows us to estimate a source contribution to each particular sample, and to investigate unique behaviors that differ from the "normal" contribution. Here, we present the NMF and VarNMF models formally, discuss algorithms to estimate parameters for both models and compare their performances on synthetic data. Finally, we apply both to real-world data, to test how VarNMF confronts the problem presented above, and what is the extent of its interpretability.

## 2    BACKGROUND AND RELATED WORKS

**Notations**    We use $i$ as sample index, and $X[i]$ as the variable X in the $i$'th sample. We use $j$ as feature index, and $k$ as component index. The input to NMF is a data matrix $V$ of $N$ samples with $M$ features, where each row is a sample $V[i] \in \mathbb{R}_{\geq 0}^M$. $V[i]_j$ denotes $j$'th entry in $V[i]$. Each sample $V[i]$ is a mix of signals from $K$ sources according to the weights $W[i] \in \mathbb{R}_{\geq 0}^K$, where each

source $k = 1, ..., K$ is represented as a constant row $H_k \in \mathbb{R}_{\geq 0}^M$, or as a sample-specific vector $H[i]_k \in \mathbb{R}_{\geq 0}^M$ sampled from a distribution $p_k$ (depending on the model, see below).

**NMF** Given a non-negative dataset $V$, the objective of NMF is to decompose the mixed signal into $K$ sources, each represented by a constant component vector $H_k \in \mathbb{R}_{\geq 0}^M$. The classic formulation from Lee & Seung (2000) is as an optimization problem (Eq. 2, left), where we look for two low-rank non-negative matrices s.t. their product is the closest to the observations, under some error function $\mathcal{D}$. An equivalent formulation of the same problem (Eq. 2, right) is as a generative model for which we try to find a maximum likelihood parameters under the assumptions of non-negativity and some noise model (Lee & Seung, 1999):

$$\hat{W}, \hat{H} = \arg\min_{W,H \geq 0} \mathcal{D}\left(V \mid\mid R\left(W, H\right)\right) \qquad \Longleftrightarrow \qquad \hat{W}, \hat{H} = \arg\max_{W,H \geq 0} P\left(V \mid W, H\right) \qquad (2)$$

$$\text{where } R\left(W, H\right) = W \cdot H \qquad\qquad\qquad \text{where } V[i]_j \sim P_{\text{obs}}\left(R[i]_j\right)$$

$$H \in \mathbb{R}_{\geq 0}^{K \times M}, W \in \mathbb{R}_{\geq 0}^{N \times K} \qquad\qquad\qquad R[i]_j = \sum_k W[i]_k \cdot H_{k,j}$$

A graphical representation of this model is given in Fig. 1D. Specifically, we will focus on the KL-NMF model, where $\mathcal{D}$ is the KL-divergence error, and its equivalent generative formulation, where the observation probability $P_{\text{obs}}$ is a Poisson distribution — a common assumption in the case of biological sequencing data (Anders & Huber, 2010).

This optimization problem is convex in $W$ and $H$ separately, but not together, thus we can only guarantee finding local minima (Lee & Seung, 2000). Many methods exist for approximating a solution. A simple and widely used method is an iterative algorithm called the *Multiplicative Update Rule* (Lee & Seung, 2000) which is equivalent to block-wise gradient descent over $W$ and $H$ separately, while choosing a learning rate which keeps all the elements of both matrices non-negative. In particular, the KL divergence is non-increasing under this multiplicative update rules and it is invariant under these updates if and only if $W$ and $H$ are at a stationary point of the divergence.

**Related Works** It is beyond our scope here to discuss the many extensions to NMF (Wang & Zhang, 2012). A common theme is adding some form of regularization, structure, or prior assumptions on the matrices $H$ and $W$ (i.e., assuming their values follow some distribution). Schmidt et al. (2009) described a Bayesian formulation of NMF, followed by many other formulations, including different choices of priors (e.g. Brouwer & Lio (2017)) and hierarchical Bayesian models (e.g. Lu & Ye (2022), Gopalan et al. (2015)). However, while these Bayesian extensions can integrate prior knowledge on the potential sources into the separation process, they still assume that constant components are shared between samples (even if these are integrated over (Brouwer & Lio, 2017)), and therefore they do not account for variation in the source signals between samples; see Fig 1.

Recently, Andrade Barbosa et al. (2021) suggested a probabilistic model in which, similarly to VarNMF, components are distributions (in their case, log-normal distributions), and therefore accounts for the variation in source signal between samples. However, they assume that these prior distributions are pre-learned, that is, estimated from direct observations of the sources. This approach can insert technical biases and fail in cases where there is no direct access to the sources, or if the sources that comprise the data are a-priori unknown. Here, we overcome this issue by estimating the sources' distributions directly from the data.

## 3   METHOD: VARNMF

**VarNMF Model** We start with the probabilistic formulation of NMF from Eq. 2, and consider the possible variation in source values between samples. To model this variation, we take each component $H_k$ to be a random vector. That is, for each sample $i$, we have the latent components matrix:

$$H[i] \in \mathbb{R}_{\geq 0}^{K \times M} \text{ s.t. } H[i]_k \overset{\text{i.i.d by } i}{\sim} p_k \qquad (3)$$

and the data distribution becomes:

$$V[i]_j \sim \text{Poisson}\left(R[i]_j\right), \text{ where } R[i]_j = \sum_k W[i]_k \cdot H[i]_{k,j} \qquad (4)$$

This model is described in its graphical form in Fig. 1D. Importantly, the component signal is now a random vector that has its own instantiation for each sample, and is sampled from the source distribution $p_k$ with some parameters $\theta_{H_k}$.

Now, given $N$ independent observation vectors $V = (V[1], ..., V[N]) \in \mathbb{R}_{\geq 0}^{N \times M}$, we look for the maximum likelihood estimator (MLE) of $\theta = (W, \theta_H)$, that is, the proportion vectors $W \in \mathbb{R}_{\geq 0}^{N \times K}$ and the source distributions' parameters $\theta_H$ s.t.

$$\hat{\theta} = \arg\max_{W, \theta_H} \mathcal{L}_{\text{VarNMF}}\left(W, \theta_H; V\right) \tag{5}$$

For simplicity, we assume that the different features in all components are independent. We also follow a common modeling choice for gene expression (Anders & Huber, 2010) and assume that for each source $k$, the signal of each feature $j$ is distributed according to a Gamma distribution with its own parameters $\theta_{H_{k,j}} = (A_{k,j}, B_{k,j})$:

$$p_k\left(H[i]_k\right) = \prod_j p_{k,j}\left(H[i]_{k,j}\right) = \prod_j p_{\text{Gamma}}\left(H[i]_{k,j}; A_{k,j}, B_{k,j}\right) \tag{6}$$

**Likelihood function**    The task defined by Eq. 5 is hard, as it involves integration over the latent matrix $H$. Specifically, computing the likelihood of a single observation requires a K-dimensional integral:

$$P\left(V[i]_j \mid \theta\right) = \int_{\vec{h} \in \mathbb{R}_{\geq 0}^K} P\left(V[i]_j \mid H[i]_{:,j} = \vec{h}, W[i]\right) \cdot p\left(H[i]_{:,j} = \vec{h} \mid A_{:,j}, B_{:,j}\right) d\vec{h} \tag{7}$$

One simplification that partially alleviate this complexity is as follows: the Poisson distribution is linear w.r.t its rate. Therefore, we can define another set of latent variables $Y$ that represent the contribution of signal from each source, with its own Poisson noise:

$$Y[i]_{k,j} \sim \text{Poisson}\left(W[i]_k \cdot H[i]_{k,j}\right) \tag{8}$$

and get the deterministic dependency

$$V[i]_j = \sum_k Y[i]_{k,j} \tag{9}$$

Essentially, we separate the Poisson noise that differentiates $V[i]_j$ from $R[i]_j$ to the noise originating from each source. Now, given the values of $Y$, the components of $H$ are independent of each other. Thus, we can replace the K-dimensional integration with a K-dimensional summation that can be calculated using dynamic programming (Appendix A.2). Moreover, $P\left(Y[i]_{k,j}; A_{k,j}, B_{k,j}\right)$ which involves integration over $H[i]_{k,j}$ has a closed form solution (Lemma A.5).

**Complete-data log-likelihood**    While we can now theoretically optimize the likelihood by searching over the parameter space, this is infeasible in practice. Instead, we use the EM procedure (Appendix A.3 for full details). We start by examining the log-likelihood as though we observe the latent variables $Y$ and $H$:

$$\ell^*\left(\theta; V, Y, H\right) \overset{\text{def}}{=} \log p\left(V, Y, H \mid \theta\right) \tag{10}$$

which can be decomposed into three factors: $\log P\left(V \mid Y\right)$, $\log P\left(Y \mid W, H\right)$ and $\log p\left(H \mid A, B\right)$. The first factor is a log delta function. The second can be further decomposed for each sample $i$ and source $k$ into a separate log-likelihood function for the parameter $w = W[i]_k$, that accounts for the Poisson noise in $Y[i]_k$:

$$\ell_{i,k}^{Y*}\left(w\right) \overset{\text{def}}{=} \log P\left(Y[i]_k \mid w, H[i]_k\right) \tag{11}$$

These likelihood functions have sufficient statistics:

$$G[i]_k \overset{\text{def}}{=} \sum_j Y[i]_{k,j}, \qquad\qquad T[i]_k \overset{\text{def}}{=} \sum_j H[i]_{k,j} \tag{12}$$

The last factor $\log p\left(H \mid A, B\right)$ represents the source distributions. Since we assumed independence between sources and between features, we can maximize the Gamma log-likelihood of each source $k$ and feature $j$ separately w.r.t. $a = A_{k,j}$ and $b = B_{k,j}$:

$$\ell_{k,j}^{H*}\left(a, b\right) \stackrel{\text{def}}{=} \log p\left(H[1]_{k,j}, ..., H[N]_{k,j} \mid a, b\right) \tag{13}$$

using the sufficient statistics of Gamma distribution:

$$S^0 \stackrel{\text{def}}{=} N, \qquad S_{k,j}^1 \stackrel{\text{def}}{=} \sum_i H[i]_{k,j}, \qquad S_{k,j}^{\log} \stackrel{\text{def}}{=} \sum_i \log H[i]_{k,j} \tag{14}$$

**Expectation Maximization (EM) procedure**    Given a starting point $\theta^{(0)} = \left(W^{(0)}, \theta_H^{(0)}\right)$, we apply the following Expectation (E-) and Maximization (M-) steps iteratively until convergence of the marginal log-likelihood $\ell_{\text{VarNMF}}\left(W, \theta_H; V\right)$:

In the E-step we calculate the expectation of the sufficient statistics (the ESS) from Eq. 12 and Eq. 14 w.r.t. the posterior $p\left(Y, H \mid V, \theta^{(t)}\right)$. The full process is described in Appendix A.3, but essentially it is sufficient to calculate for each sample $i$ and feature $j$:

$$\forall_{k,d}, \ p\left(V[i]_j \mid Y[i]_{k,j} = d; \theta^{(t)}\right) = p\left(\sum_{l \neq k} Y[i]_{l,j} = V[i]_j - d; \theta^{(t)}\right) \tag{15}$$

which can be achieved using the same dynamic programming procedure of the log-likelihood.

In the M-step, we maximize the expectation of the complete-data log-likelihood:

$$\theta^{(t+1)} = \arg\max_\theta \mathbb{E}_{p\left(Y,H \mid V, \theta^{(t)}\right)}\left[\ell^*\left(\theta; V, Y, H\right)\right] \tag{16}$$

From linearity of expectation, we can find $\theta^{(t+1)} = (W, A, B)$ by separately maximizing

$$\mathbb{E}\left[\ell_{i,k}^{Y*}\left(w\right)\right] \qquad\qquad \mathbb{E}\left[\ell_{k,j}^{H*}\left(a, b\right)\right] \tag{17}$$

These are the same functions as the log-likelihood functions $\ell_{i,k}^{Y*}$ and $\ell_{k,j}^{H*}$, only with the ESS calculated in the E-step replacing the actual sufficient statistics. Therefore, they can be maximized in the same way.

**Convergence and implementation**    Following the EM scheme, we assure convergence to a local maximum. In our case, the E-step is computationally demanding while the M-step is straightforward. As a starting point we use the NMF solution of $W$ and $H$. We use the estimated $H$ to initialize the mean of the Gamma distributions over $H$ and initialize the variance with a constant coefficient of variation. We use a simple stopping criteria of $T = 250$ iterations. An additional issue with both NMF and VarNMF solutions is that they are not identifiable, and many solutions will have the same log-likelihood. However, in practice, NMF will find solution that is unique up to permutation and normalization (Huang et al., 2013). Therefore, we use a simple normalization scheme to scale NMF and VarNMF solutions (Appendix A.6).

**Posterior expected signal**    Using the training data, we estimate a prior distribution for each component, $\hat{p}_k = \text{Gamma}\left(A_k, B_k\right)$. The mean of this distribution can be interpreted similarly to the constant components provided by NMF. However, under the VarNMF model assumptions, each source $k$ contributes some signal $H[i]_k$ to sample $i$, weighted by $W[i]_k$. This sample-specific signal represents the isolated contribution of source $k$ to sample $i$, and estimating it can help identify cases where the true contribution of the source to the particular sample is far from expected. We estimate this signal using the expectation of the *posterior source distribution* of the sample,

$$\hat{p}[i]_k\left(h\right) = p\left(H[i]_k = h \mid V[i], A_k, B_k\right) \tag{18}$$

which is calculated during the E-step (Appendix A.4).

## 4    EXPERIMENTS

### 4.1    CELL-FREE CHIP-SEQUENCING

Our motivation stems from analysis of genomic data from an assay called *cell-free chromatin immunoprecipitation followed by sequencing* (cfChIP-seq) (Sadeh et al., 2021) (Appendix C). This

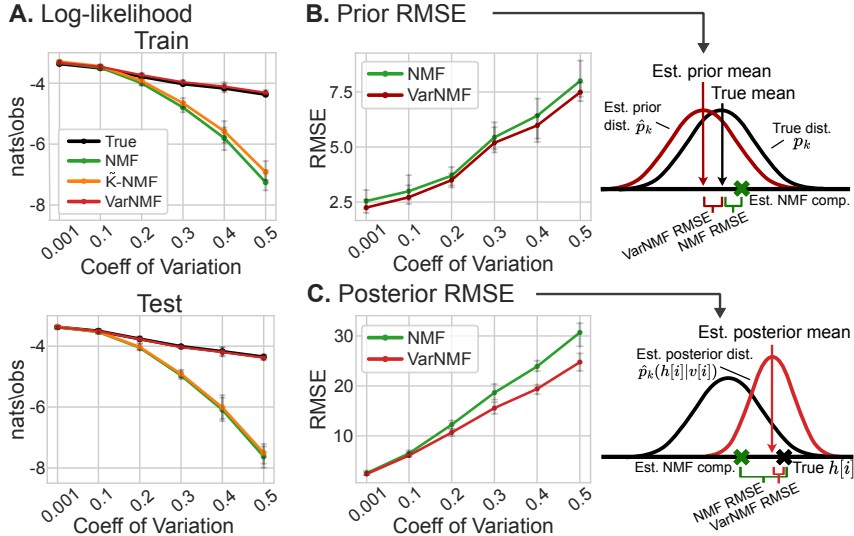

Figure 2: Decomposing synthetic data with $K = 4$ components: A) Train and test log-likelihood of the ground truth parameters and three models - NMF, VarNMF and $\tilde{K}$-NMF (NMF with higher degrees of freedom than VarNMF), versus the coefficient of variation of the dataset. The log-likelihood values are normalized to nats/observation. B) Root mean square error (RMSE) of the mean of the distributions estimated by VarNMF (red on right panel) and the constant components estimated by NMF (green) versus the mean of the ground truth distributions (black). C) RMSE of the per-sample posterior expected signal estimated by VarNMF (red) and the constant components estimated by NMF (green) versus the ground truth $H[i]$ (black). A,B,C show results for $T = 10$ runs.

assay is performed on plasma (the cell-free portion of blood) and captures DNA fragments marked with a specific epigenetic modification called H3K4me3. This cell-free DNA in the plasma originates from dying cells of various sources — cell-types or tissues across the body — according to different mixing weights that depends on the cell-type death proportions in the individual. The chosen modification marks DNA fragments that were located in active genes in the original cells (Soares et al., 2017). Therefore, the signal from each of these cell-types is tightly coordinated with gene activity. Decomposing the mixed signal into individual components would identify gene related activity in each sub-population of cells contributing to it (e.g., tumor cells, immune cells). This is crucial for interpreting complex samples.

Sadeh et al. (2021) applied cfChIP-seq on large number of subjects. Their results display differences between samples in the signal originating from specific cell-types, which are not explained by the proportions of cell-death. For example, in a cohort of patients with liver diseases, there are clear differences between the patient samples and the healthy baseline (Fig. 5 of Sadeh et al.). Accounting for the increased cell-death proportions of liver cells in the patients accounts for some of these differences, but not all of them, even when focusing on genes that are specific to the liver. This suggests variation between samples in the liver component signal. In theory, one can assay many liver biopsies of different pathological states to estimate the liver component distribution. However, in practice, this is fraught with logistical and financial difficulties. For other tissue types (e.g., heart, brain) this is essentially impossible. Therefore, this data present an interesting use-case for our method. We start by applying VarNMF to synthetic data based on a cfChIP-seq dataset properties with increasing variation in the components signal, and compare the results with those of the NMF model. Next, we apply both algorithms to a cfChIP-seq dataset, to see whether VarNMF can better explain this data.

## 4.2 SYNTHETIC DATA

To illustrate the capability of VarNMF for non-negative decomposition with source variation, we consider mixed datasets with $M = 100$ features and $K = 1, ..., 10$ sources. The sources are modeled

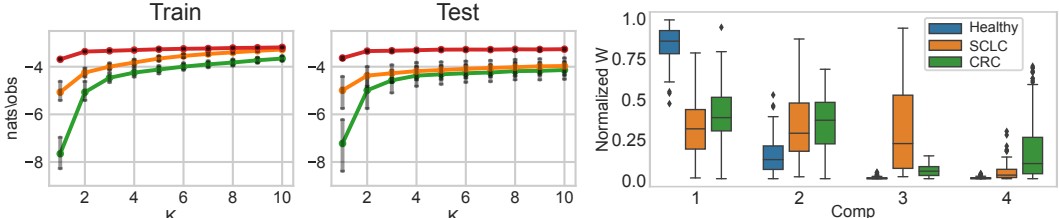

Figure 3: Decomposing cell-free ChIP-seq data: A) Train and test log-likelihood curves for real cfChIP-seq dataset versus the number of components $K$ used, for the three models and for $T = 5$ splits to train and test. The log-likelihood values are normalized to nats/observation. B) The VarNMF estimated weights for each component in the $K = 4$ solution (train and test shown together), aggregated by sample sub-cohort (healthy, SCLC and CRC). Weights are normalized so that each sample has a total weight of 1.

as Gamma distributions with mean $\mu_{k,j}$ and variance $\sigma_{k,j}^2$ for the $j$'th feature in the $k$'th component. The means are generated by random permutations of an NMF solution trained on a real cfChIP-seq dataset. We control the variation level of the sources by setting the coefficient of variation to a constant value, thus $\sigma_{k,j} = \frac{\text{CV}}{\mu_{k,j}}$. In each realization, we generate $N = 100$ samples from each source $H[i]_{k,j} \overset{\text{i.i.d.}}{\sim} \text{Gamma}\left(A_{k,j}, B_{k,j}\right)$ with $A_{k,j} = \frac{1}{\text{CV}^2}, B_{k,j} = \frac{1}{\text{CV}^2 \mu_{k,j}}$ (Lemma A.2), and mix them using a weight matrix $W$ whose rows are drawn i.i.d. from a symmetric Dirichlet $(1)$ distribution, multiplied by a per-sample scaling factor $\lambda[i]$ drawn from a uniform distribution $\mathcal{U}\left([1, 2]\right)$. We then sample an observation $V[i]_j$ from a Poisson with the mixture $R[i]_j = W[i] \cdot H[i]_j$ as its rate.

To evaluate VarNMF vs. NMF, we examine their performance with the correct number of sources $K$. Since the number of parameters in VarNMF is higher than in NMF, we also apply NMF with $\tilde{K}$ sources, where $\tilde{K}$ is the minimal number to compensate for the difference in degrees of freedom.

Our goal is to extract accurate source distributions to better analyse new samples. Thus, we measure the generalization abilities of each model on a new dataset sampled from the same distribution. We created test datasets with $N_{\text{test}} = 100$ and used a version of NMF and VarNMF to fit new weight matrices $W_{\text{test}}$ while keeping the learned components\ source distributions constant (Appendix A.7).

Applied to the synthetic datasets, the proposed VarNMF model achieves high log-likelihood for the train data, despite increasing CV (Fig. 2A and Appendix B.1 ). Although the NMF curve has similar values for low levels of variation, it drops sharply when these levels increase. This is not due to differences in number of parameters, as $\tilde{K}$-NMF follows the same trend. Importantly, the test log-likelihood performance of VarNMF is similar to the train results. The results are also similar when increasing the number of samples $N$ and the observational noise $\lambda$ (Appendix B.2). This suggests that VarNMF better captures the datasets distribution in the presence of high component variation.

Next, we examine the learned components against the ground truth parameters and observe that the means of VarNMF distributions are closer to the ground truth means than NMF constant estimates (Fig. 2B and Appendix B.1 ). A similar improvement can be seen when comparing the ground truth per-sample contribution of source $k$, $H[i]_k$, to the VarNMF posterior expected signal versus the NMF constant components (Fig. 2C and Appendix B.1 ).

Overall, we conclude that for data with component variation, VarNMF learns more accurate mean signal for the different components, and allows for a sample-specific source value estimation via the posterior expect signal.

### 4.3 REAL DATA

We collected a dataset of cfChIP-seq samples from Sadeh et al. (2021) and Fialkoff et al. (2022). This data includes plasma samples of 80 healthy subjects, 139 small-cell lung cancer (SCLC) patients, and 86 colorectal cancer (CRC) patients. The two cancer cohorts represent different diseases,

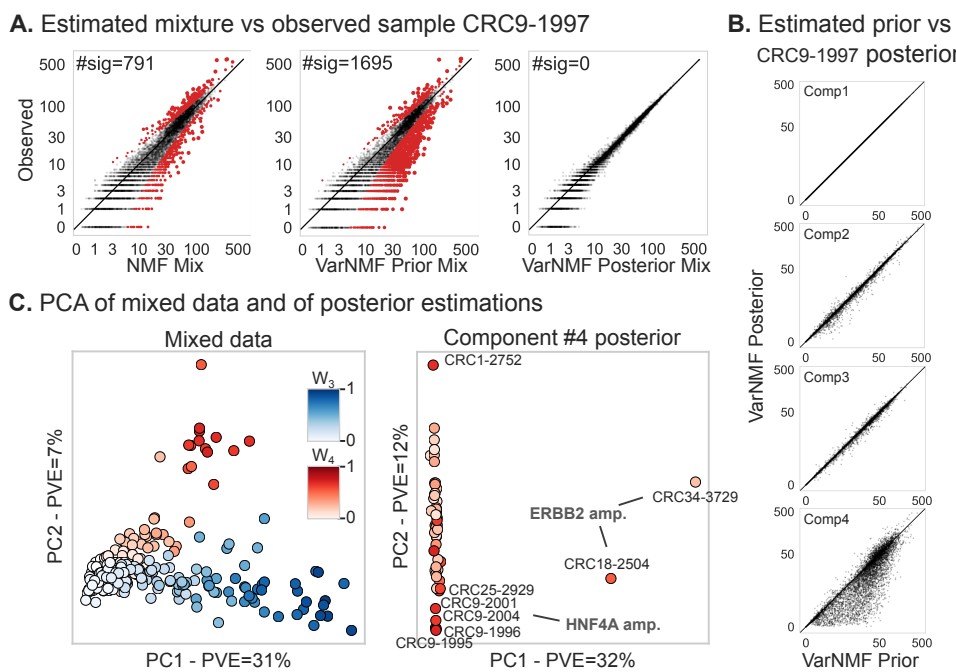

Figure 4: Reconstructing $H$ from data with the $K = 4$ solution. A) Reconstruction quality of a specific sample by NMF (left), VarNMF mean components (middle), and VarNMF posterior (right). Each point is a gene, the x-axis shows the reconstructed value and y-axis the observed value. Red points are ones that are significantly different, taking into account Poisson sampling noise (q-values corrected for false discovery rate - $< 0.05$). B) For the same example, the prior (component mean) vs. posterior estimate per component. C) Principal Component Analysis (PCA) of the original mixed samples after normalization (Appendix C) (left) and component-wise reconstruction (right).

yet they are both solid tumors and are expected to have some common features. As noted by Sadeh et al., patient samples vary in the amount of tumor contribution to the plasma (e.g., lower contribution following treatment). Thus, there is non-trivial heterogeneity in terms of the mixing proportions. They also report on molecular differences among cancers of the same type, and therefore we expect to observe some variability in the signal of the component(s) representing each cancer type.

Out of this dataset, we selected $M = 7000$ features (Appendix C.1). Additionally, instead of training with the EM algorithm directly, we use a scheme that alternates optimization of $W$ and $A, B$, to allow for parallel optimization of the Gamma distributions of the different features (Appendix A.5). Another adjustment for this data is for a non-specific background noise originating from the sequencing assay (Sadeh et al., 2021). The levels of this noise are estimated as part of the assay and we regard to it as another source of signal (i.e., another component), with pre-estimated contribution and signal. In particular, both NMF and VarNMF can incorporate this noise with minor changes to their training algorithms. We randomly divided the cohort into 185 training samples and 120 test samples (repeated 5 times). We trained the different procedures on the training data with increasing number of components $K$ without any labels. We test the trained model using the same scheme as for synthetic data (Appendix A.7).

We start by evaluating the ability of the different models to generalize to new instances from the same cohort. Plotting the log-likelihood as a function of the number of components $K$ (Fig. 3A), we see that while all models perform better on training data when $K$ increases, there is a clear advantage for VarNMF on test data ($\sim 1.2$ nats/observation) that does not decrease as with larger $K$s. We thus conclude that VarNMF learns better representations of the inherent structure in the data.

Next, we examine the biological relevance of the learned components in a specific solution. While there is no clear optimal $K$, we choose the $K = 4$ solution as a representative example (Appendix C.3 for analysis of other values of $K$). Examining the range of values of $W$ for the three sub-

cohorts (Fig. 3B) we see two cancer-specific components (that have non-zero weights mostly in one cancer type) - components #3 and #4, and two shared "healthy" components (that are high in healthy samples but also appear in a range of weights in the cancer samples) - components #1 and #2. This is to be expected given that the cancer contributes only a part of the cell-free material in the plasma.

An alternative way of interpreting the biological association of the components is to examine the mean contribution of a component to the value of each gene. Specifically, we can choose genes that have high mean value in the component and low mean values in all other components (Appendix C.2). We then test whether these genes are significantly over-represented in curated genes-lists associated with a specific tissue or cell-type (Chen et al., 2013). We observe that the two cancer-specific components are associated with the cell-types effected in each disease. In contrast, the two "healthy" components specific genes are enriched for two different types of blood related (hematopoietic) cells, which were identified by Sadeh et al. (2021) as the main sources of cell-free DNA in healthy donors (Appendix C.2). We conclude that the mean of the components estimated by VarNMF captures the main expected contributors of cell-free DNA to the cohort samples. Importantly, the NMF solution yields a similar interpretation, and these findings are not exclusive to the VarNMF estimation (Appendix C.2 and C.3).

To illustrate the unique features of VarNMF, we examine a specific sample of a CRC-patient (Fig. 4A). Mixing the NMF constant components according to the weights learned for this sample results in a reconstruction that significantly diverges from the observed signal in hundreds of genes. Similarly, mixing the means of the component distributions learned by VarNMF according to the VarNMF weights for the sample also results in many unexplained genes, even more than for the NMF reconstruction. However, when using the sample-specific component posteriors estimated by VarNMF, the observed signal is fully explained. Thus, while much of the variation between samples can be accounted for by the mixing proportions, there is a non-trivial additional layer of variability due to variation in signals within components. Examining these posterior signals (Fig. 4B), we notice that while for the first three components the posterior signal is close to the prior value, the fourth CRC-associated component exhibits sample-specific signal. This suggest that most of the discrepancies originate from the CRC source, i.e., that the disease-specific features explain most of the previously unexplained signal in the sample. These behaviors are repeated in most samples in both the train and test datasets.

Looking at this phenomena more generally (Fig. 4C), the main directions of variation in the mixed samples are the estimated percentage of disease (Pearson correlation of PC1 and $W_3 = 0.89$; of PC2 and $W_4 = 0.84$). In contrast, the posterior expected signal allows us to separate sample-specific disease behavior and observe inter-cancer variability: the main directions of variation are no longer associated with degree of disease, but with unique features of sub-populations of patients. For example, in the CRC-associated component (component #4), the first two PCs of the posterior expected signal separate two sub-populations from the main population. These sub-populations have two different known genomic amplifications (Sadeh et al., 2021) (ERBB2 with PVE=32% and HNF4A with PVE=12%). Importantly, degree of disease is no longer a main contributor of variation.

## 5  CONCLUSION

Here, we presented VarNMF, a method for decomposition of non-negative sequencing data into source distributions and mixing proportions, in a way that tackles variation in source values between samples. We show that VarNMF can decompose real-world sequencing data into biologically relevant source distributions that represent the entire population. Additionally, VarNMF has the potential to extract the sample-specific posterior expected signal of a source, that reflects the source value in the specific sample, and can indicate patient-specific disease behavior. While the alternating EM procedure allows to scale to large dataset, it remains computationally expensive and further efforts may be taken to speed up training.

In a broader perspective, the approach we presented here provides a framework for learning about distributions of complex latent sources without observing them directly. In principle the ideas we presented here can be generalized for learning more involved models of $p(H_k)$, for example, one with some dependencies between features. This requires handling the computational challenges of computing the E-step of the EM procedure. It also raises interesting questions regarding the limitations of generalizing such models from mixed observations.

## REPRODUCABILITY STATEMENT

cfChIP data is based on Sadeh et al. (2021) [publicly available data] and Fialkoff et al. (2022) [provided through personal communications with authors]. The real data matrix and the code to reproduce the experiments is available in the supplementary file.

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

## A EM DETAILS

### A.1 DISTRIBUTIONS DEFINITIONS AND PROPERTIES

We use the following parametrizations of the Poisson, Gamma and NB distributions:

**Definition A.1.** (Poisson Parametrization) We say that $Y \sim \text{Poisson}(\lambda)$ if

$$P(Y = k) = \frac{1}{k!} \cdot \lambda^k \cdot e^{-\lambda}$$

**Lemma A.1.** (Poisson Properties) For $Y \sim \text{Poisson}(\lambda)$,

$$\mathbb{E}[Y] = Y, \quad \text{Var}[Y] = Y, \quad \text{CV}[Y] = 1/\sqrt{\lambda}$$

**Definition A.2.** (Gamma Parametrization) We say that $X \sim \text{Gamma}(\alpha, \beta)$ if

$$p(X = x) = \frac{\beta^\alpha}{\Gamma(\alpha)} \cdot x^{\alpha-1} \cdot e^{-\beta x}$$

**Lemma A.2.** (Gamma Properties)

1. For $X \sim \text{Gamma}(\alpha, \beta)$,

$$\mathbb{E}[X] = \frac{\alpha}{\beta}, \quad \text{Var}[Y] = \frac{\alpha}{\beta^2}, \quad \text{CV}[Y] = 1/\sqrt{\alpha}$$

2. If

$$\mathbb{E}[X] = \mu, \quad \text{Var}(Y) = \sigma^2$$

then

$$X \sim \text{Gamma}\left(\alpha = \frac{\mu^2}{\sigma^2}, \beta = \frac{\mu}{\sigma^2}\right)$$

3. If

$$\mathbb{E}[X] = \mu, \quad \text{CV}[Y] = \text{cv}$$

then

$$X \sim \text{Gamma}\left(\alpha = \frac{1}{\text{cv}^2}, \beta = \frac{1}{\mu \cdot \text{cv}^2}\right)$$

**Definition A.3.** (Negative Binomial Parametrization) We say that $X \sim \text{NB}(r, p)$ if

$$P(X = k) = \binom{k + r - 1}{k} \cdot p^k \cdot (1 - p)^r$$

**Lemma A.3.** (Negative Binomial Properties) For $X \sim \text{NB}(r, p)$,

$$\mathbb{E}[X] = \frac{pr}{1 - p}, \quad \text{Var}[X] = \frac{pr}{(1 - p)^2}, \quad \text{CV}[Y] = 1/\sqrt{pr}$$

and state lemmas concerning properties of the Gamma and Poisson distributions (Wackerly et al., 2014):

**Lemma A.4.** If $X \sim \text{Gamma}(\alpha, \beta), \gamma > 0$ then $\gamma \cdot X \sim \text{Gamma}\left(\alpha, \frac{\beta}{\gamma}\right)$.

**Lemma A.5.** If $Y \mid X \sim \text{Poisson}(X)$ and $X \sim \text{Gamma}(\alpha, \beta)$ then $Y \sim \text{NB}\left(\alpha, \frac{1}{1+\beta}\right)$.

**Lemma A.6.** If $Y \mid X \sim \text{Poisson}(\gamma X)$ and $X \sim \text{Gamma}(\alpha, \beta)$ then

$$(X|Y = t) \sim \text{Gamma}(\alpha + t, \beta + \gamma)$$

**Lemma A.7.** If $X \sim \text{Gamma}(\alpha, \beta)$ then $\mathbb{E}[\log(X)] = \psi(\alpha) - \log \beta$, where $\psi$ is the digamma function.

### A.2 LOG-LIKELIHOOD

First, we present a scheme that allow us to calculate the log-likelihood of a set of parameters $\theta = W, A, B$. We can calculate the log-likelihood for each observation separately:

$$\ell(\theta; V) \stackrel{\text{def}}{=} \log \sum_{i,j} p\left(V[i]_j \mid W[i], A_{:,j}, B_{:,j}\right) \tag{19}$$

To simplify notation, we set $i, j$ and define

$$v = V[i]_j, y = Y[i]_{:,j}, h = H[i]_{:,j}, a = A_{:,j}, b = B_{:,j}, w = W[i], \theta = (w, a, b) \tag{20}$$

and look to calculate $\log p(v \mid \theta)$. As mentioned above, with the addition of the random variables $Y$ to the model, this require a K-dimensional summation, and will be done using dynamic programming.

It is sufficient to calculate the joint distribution of $(y_k, v)$ for some $k$, since:

$$p(v \mid \theta) = \sum_{d=0}^{v} p(y_k = d, v \mid \theta) \tag{21}$$

Moreover, we can write this joint as

$$p(y_k = d, v \mid \theta) = p(y_k = d \mid \theta) \cdot p(v \mid y_k = d, \theta) \tag{22}$$

The first factor can be directly calculated using the following lemma:

**Lemma A.8.**

$$\begin{cases} (y_k \mid w_k, h_k) \sim \text{Poisson}(w_k \cdot h_k) \\ (h_k \mid a_k, b_k) \sim \text{Gamma}(a_k, b_k) \end{cases} \Rightarrow (y_k \mid w_k, a_k, b_k) \sim \text{NB}\left(a_k, \frac{w_k}{w_k + b_k}\right)$$

*Proof.* From Lemma A.4,

$$(h_k \mid a_k, b_k) \sim \text{Gamma}(a_k, b_k) \Rightarrow (w_k \cdot h_k \mid w_k, a_k, b_k) \sim \text{Gamma}\left(a_k, \frac{b_k}{w_k}\right)$$

and from Lemma A.5 we get

$$\begin{cases} (y_k \mid w_k, h_k) \sim \text{Poisson}(w_k \cdot h_k) \\ (w_k \cdot h_k \mid w_k, a_k, b_k) \sim \text{Gamma}\left(a_k, \frac{b_k}{w_k}\right) \end{cases}$$

and therefore

$$(y_k \mid w_k, a_k, b_k) \sim \text{NB}\left(a_k, \frac{1}{1 + \frac{b_k}{w_k}}\right) = \text{NB}\left(a_k, \frac{w_k}{w_k + b_k}\right)$$

$\square$

As for the second factor, $v = \sum_k y_k$ and $v$ is discrete, therefore

$$\forall_{k,d}, \; p\left(v \mid y_k = d, \theta\right) = p\left(\sum_{l \neq k} y_l = v - d \mid \theta\right) \tag{23}$$

and that can be calculated using dynamic programming:

For simplicity, we denote $p = p_\theta$ and $p_k = \mathrm{NB}\left(a_k, \frac{w_k}{w_k + b_k}\right)$ (the distribution of $y_k$). We define two random variables:

$$X_s \overset{\text{def}}{=} \sum_{l=1}^{s} y_l \qquad\qquad Z_s \overset{\text{def}}{=} \sum_{l=s+1}^{K} y_l \tag{24}$$

and two tables

$$F\left[s, n\right] \overset{\text{def}}{=} p\left(X_s = n\right), \quad s = 1, ..., K - 1, \; n = 0, ..., v \tag{25}$$

$$B\left[s, n\right] \overset{\text{def}}{=} p\left(Z_s = n\right), \quad s = 1, ..., K - 1, \; n = 0, ..., v \tag{26}$$

Using law of total probability, we get the two following recursive formulas:

$$F\left[s, n\right] = \sum_{d=0}^{n} F\left[s - 1, d\right] \cdot p_s\left(n - d\right) \tag{27}$$

$$B\left[s, n\right] = \sum_{d=0}^{n} B\left[s + 1, d\right] \cdot p_{s+1}\left(n - d\right) \tag{28}$$

Now we have two dynamic programming tasks:

1. The Forward task of filling $F$ by columns, from the initialization:

$$\forall_{0 \leq n \leq v}, \; F\left[1, n\right] = p_1\left(n\right)$$

   and forward according to Eq. 27.

2. The Backward task of filling $B$ by columns, with initial values for the last column:

$$\forall_{0 \leq n \leq v}, \; B\left[K - 1, n\right] = p_K\left(n\right)$$

   and going backward with Eq. 28.

Finally, in order to find the required probability we use these two tables to fill a new table

$$P\left[k, d\right] \overset{\text{def}}{=} p\left(v \mid y_k = d\right), \quad k = 1, ..., K, \; d = 0, ..., v \tag{29}$$

using the recursive formula

$$\forall_{1 < k < K}, \forall_d, \quad P\left[k, d\right] = p\left(\sum_{l \neq k} y_l = v - d\right) \tag{30}$$

$$= p\left(\sum_{l=0}^{k-1} y_l + \sum_{l=k+1}^{K} y_l = v - d\right)$$

$$= p\left(X_{k-1} + Z_k = v - d\right)$$

$$= \sum_{n=0}^{v-d} p\left(Z_k = v - d - n\right) \cdot p\left(X_{k-1} = n\right)$$

$$= \sum_{n=0}^{v-d} B\left[k, v - d - n\right] \cdot F\left[k - 1, n\right]$$

and the initial condition

$$\forall_d, \; P\left[K, d\right] = F\left[K - 1, v - d\right] \tag{31}$$

### A.3 COMPLETE-DATA LOG-LIKELIHOOD

To apply the EM procedure to our model, we need to calculate the expectations of the following sufficient statistics to get the ESS in the E-step:

$$G[i]_k = \sum_j Y[i]_{k,j}, \qquad\qquad T[i]_k = \sum_j H[i]_{k,j} \qquad (32)$$

$$S^0 = N, \qquad S^1_{k,j} = \sum_i H[i]_{k,j}, \qquad S^{\log}_{k,j} = \sum_i \log H[i]_{k,j} \qquad (33)$$

and to maximize the expectation of the complete-data log-likelihood in the M-step:

$$\theta^{(t+1)} = \arg\max_\theta \mathbb{E}_{p\left(Y,H|V,\theta^{(t)}\right)}\left[\ell^*\left(\theta; V, Y, H\right)\right] \qquad (34)$$

Here, we provide the full mathematical details of these steps:

**M - Step** As mentioned above, the maximization of Eq. 34 can be achieved by separately maximizing the log-likelihood functions $\ell^{Y*}_{i,k}$ and $\ell^{H*}_{k,j}$, only with the ESS calculated in the E-step replacing the actual sufficient statistics.

Specifically, $Y[i]_{k,j} \sim \text{Poisson}\left(W[i]_k \cdot H[i]_{k,j}\right)$, therefore for each $w = W[i]_k$,

$$\ell^{Y*}_{i,k}(w) \stackrel{\text{def}}{=} \log P\left(Y[i]_k \mid w, H[i]_k\right) \qquad (35)$$
$$= \text{const} + G[i]_k \cdot \log w - T[i]_k \cdot w$$

Additionally, $H[i]_{k,j} \sim \text{Gamma}\left(A_{k,j}, B_{k,j}\right)$ and thus for each $a = A_{k,j}, b = B_{k,j}$,

$$\ell^{H*}_{k,j}(a,b) \stackrel{\text{def}}{=} \log p\left(H[1]_{k,j}, ..., H[N]_{k,j} \mid a, b\right) \qquad (36)$$
$$= [a \log b - \log \Gamma(a)] \cdot S^0 - b \cdot S^1_{k,j} + (a-1) \cdot S^{\log}_{k,j}$$

Now, given the ESS, we maximize both functions separately. Set $i, j, k$, and

$$G = G[i]_{k,j}, \qquad T = T[i]_{k,j}, \qquad S^1 = S^1_{k,j}, \qquad S^{\log} = S^{\log}_{k,j} \qquad (37)$$

Then, differentiating each function and setting the gradient to 0,

1. For $\ell^{Y*}$ we get that $\hat{w} = \frac{G}{T}$.

2. For $\ell^{H*}$, we get the following system,

$$\begin{cases} \psi(a) - \frac{S^{\log}}{S^0} - \log a + \log \frac{S^1}{S^0} = 0 \\ b = a \cdot \frac{S^0}{S^1} \end{cases}$$

where $\psi(x) = \frac{(\Gamma(x))'}{\Gamma(x)}$ is the digamma function. The first equation can be solved by finding a root using the Newton Raphson algorithm.

**E - Step** As described above, given $\theta^{(t)} = W, A, B$, we are required to calculate the expectation of the sufficient statistics from Eq. 32 and Eq. 33. To simplify notation, and as this process is done separately for each feature $j$ in each sample $i$, we set $i, j$ and define

$$v = V[i]_j, y = Y[i]_{:,j}, h = H[i]_{:,j}, a = A_{:,j}, b = B_{:,j}, w = W[i], \theta^{(t)} = (w, a, b) \qquad (38)$$

From linearity of expectation, it is enough to calculate for each k,

$$\text{(i) } \mathbb{E}\left[y_k \mid v, \theta^{(t)}\right] \qquad \text{(ii) } \mathbb{E}\left[h_k \mid v, \theta^{(t)}\right] \qquad \text{(iii) } \mathbb{E}\left[\log h_k \mid v, \theta^{(t)}\right] \qquad (39)$$

We start by calculating the posterior distribution of $y_k$, $p(y_k = d \mid v, \theta)$ for each $0 \le d \le v$. This can be done using the joint probability $p(y_k = d, v \mid \theta)$ calculated with the dynamic programming scheme from A.2, and the fact that

$$p(y_k = d \mid v, \theta) = \frac{p(y_k = d, v \mid \theta)}{p(v \mid \theta)} \tag{40}$$
$$= \frac{p(y_k = d, v \mid \theta)}{\sum_{l=0}^{v} p(y_k = l, v \mid \theta)}$$

Given the posteriors, $\{p(y_k = d \mid v, \theta)\}_{d=1}^{v}$, we can now calculate:

1. $\mathbb{E}[y_k \mid v, \theta] = \sum_{d=0}^{v} d \cdot p(y_k = d \mid v, \theta)$.

2. $\mathbb{E}[h_k \mid v, \theta] = \frac{a_k + \mathbb{E}[y_k \mid v, \theta]}{b_k + w_k}$ (Lemma A.6).

3. $\mathbb{E}[\log h_k \mid v, \theta] = -\log(b_k + w_k) + \sum_{d=0}^{v} p(y_k = d \mid v, \theta) \cdot \psi(a_k + d)$
   (Lemmas A.6, A.7).

### A.4 CALCULATING THE POSTERIOR EXPECTED SIGNAL

Given a prior source distribution (for some feature $j$) $p_{k,j} = \text{Gamma}(A_{k,j}, B_{k,j})$ estimated from training data, we want to estimate the source contribution to a particular sample $i$, $H[i]_{k,j}$. We use the expectation of the posterior distribution of $H$ according to the estimated prior distribution:

$$\hat{p}[i]_{k,j}(h) = p(H[i]_{k,j} = h \mid V[i], A_{k,j}, B_{k,j}) \tag{41}$$

Specifically, we use $\mathbb{E}\left[H[i]_{k,j} \mid V[i], A_{k,j}, B_{k,j}\right]$ which we calculated in the E-step (Eq. 39(ii)).

### A.5 ALTERNATING EM

As mentioned above, the E-step of the presented EM procedure is computationally heavy, and cannot be applied on a real dataset with a typical number of a few thousand features (genes). There are many possible solutions to this problem in literature. Here, we decided to use an alternating version of EM (Neal & Hinton, 1998), which also alternates between optimizing W and A,B (similar to the NMF multiplicative update rule). The algorithm steps are detailed in Fig. 5.

We start in step 1 with an NMF solution for $W^{\text{NMF}}$ (which can be calculated fairly quickly, using the NMF algorithm mentioned above). For step 2, we use a variant of the EM procedure we call EMcW, to estimate $(A, B)$ while keeping $W^{\text{NMF}}$ constant. This process requires the calculation of the ESS of $S^1, S^{\log}$ for the E-step, and the maximization of $(A, B)$ for the M-step. We note that given $W^{\text{NMF}}$, the parameters $(A, B)$ and the ESS of $S^1, S^{\log}$ are independent between features. Thus we can divide the features into batches and parallelize the EMcW procedure entirely.

Next, we use the resulting $(A, B)$ parameters to determine which features are component-specific: We take $\left(\tilde{A}, \tilde{B}\right)$ to be the resulting $(A, B)$ for the $\tilde{M} = 100 \cdot K$ features with the highest difference between the mean signal of different components (normalized by mean signal of the features). We then use these features in step 3 to adjust the starting point $W^{\text{NMF}}$, by running the EM algorithm to estimate the mixing weights W while keeping $\left(\tilde{A}, \tilde{B}\right)$ constant (EMcAB variant). This requires the calculation of the ESS of $G, T$ in the E-step, and the maximization of $W$ in the M-step, and results in $\hat{W}$ that is adjusted for component variation. Lastly, in step 4 we readjust the component parameters $\left(\hat{A}, \hat{B}\right)$ using the EMcW variant with the constant $\hat{W}$.

In our case we stop here and return the resulting $\hat{W}$ and $\left(\hat{A}, \hat{B}\right)$ from the last step. However, this alternation can clearly be repeated several times (steps 3 and 4). Since the EMcW variant is parallelized by batches and the EMcAB only uses $\tilde{M} = 100 \cdot K$ features, each iteration is much more efficient in both time and memory than running the regular EM procedure over a large number of features.

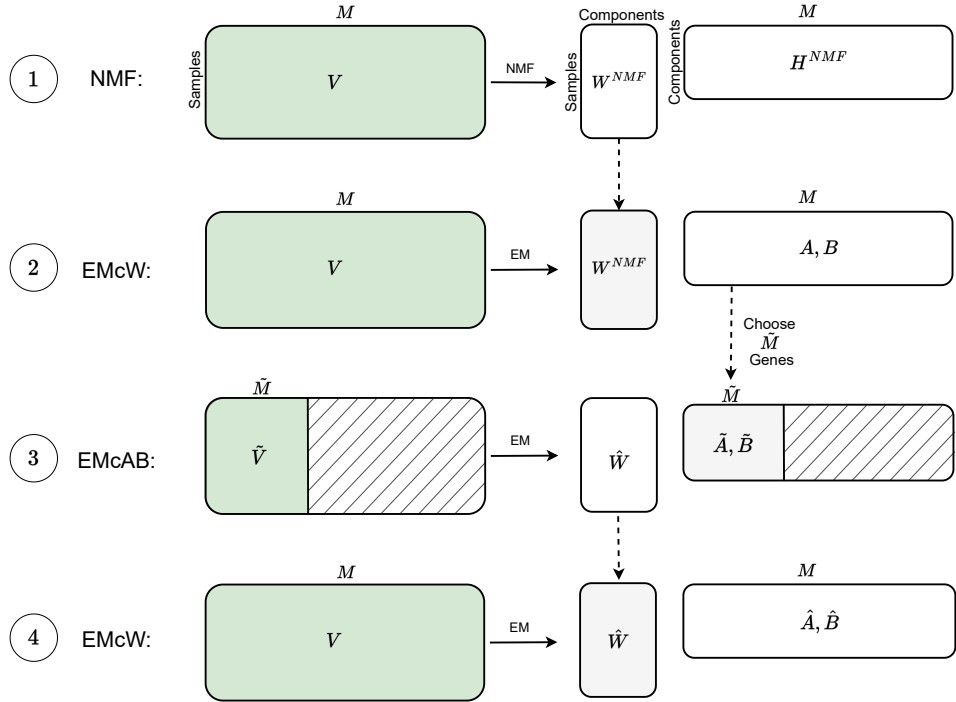

Figure 5: The alternating EM procedure, which attempts to find an optimal solution for the VarNMF model while avoiding high computational cost. It uses the NMF solution as a starting point (step 1), and apply EM over one of the parameter matrices $W$ or $(A, B)$ at a time (steps 2 and 3). This process is then repeated once in step 4 for $W$, and can be applied iteratively. Green indicates a data matrix (or a subset of one) and grey indicates a fixed parameter matrix (that results from the previous step).

## A.6 NORMALIZATION

A known limitation of the NMF model is that it is not identifiable: Say $W, H$ are an NMF solution for dataset $V$. Then, for every invertible matrix $J$ s.t.

$$W^{\text{new}} \stackrel{\text{def}}{=} W \cdot J \geq 0 \qquad\qquad H^{\text{new}} \stackrel{\text{def}}{=} J^{-1} \cdot H \geq 0 \qquad\qquad (42)$$

we have

$$W^{\text{new}} \cdot H^{\text{new}} = (W \cdot J) \cdot \left(J^{-1} \cdot H\right) = W \cdot H \qquad\qquad (43)$$

As the Poisson log-likelihood NMF depends only on the reconstruction $R = WH$ (Eq. 2), the cost of the two solutions will be identical, and therefore the KL-NMF problem is not identifiable.

This is important in a few ways: First, measuring the quality of a solution, we want to compare the solution directly to ground truth parameters (if exist). We also compare two different solutions from different models. More crucially, we want, for example, to be able to infer component patterns and use them to characterize states in the system (e.g. cell-types). If the components are not unique in a way that changes the real-life interpretation of them, it significantly harms the interpretability of the model.

We start by considering permutation and non-negative normalization matrices (diagonal matrices with non-negative elements). Since such matrices are invertible and their inverse is also non-negative (Plemmons & Cline, 1972), multiplying an NMF solution by them (as detailed in Eq. 43) will result in an equivalent solution. However, we can easily normalize NMF solutions to alleviate ambiguity of normalization and permutation and get *essentially unique* solutions that are comparable between models and runs: We use a reference solution to determine the order of the components (for synthetic data - according to the ground truth, for real data - some predefined order), and match the components order of the new solution using linear sum assignment. We then reorder the components in

the solutions by multiplying $H$ and $W$ with a corresponding permutation matrix. For normalization, we normalize each component by a factor $d_k$ by multiplying $H$ and $W$ with a corresponding normalization matrix. This results in an equivalent solution.

For VarNMF solutions we use the same process, except for the normalization of the components that requires a different treatment: We want to scale the distributions of $H$ by some factors $d_1, ..., d_K$. Following Lemma A.4, we can simply scale the parameters $B_k$ by the inverse $d_k^{-1}$, or equivalently multiply $B$ by the inverse of the corresponding normalization matrix. This will result in an equivalent solution if we normalize $W$ accordingly.

The choice of normalization depends on the data. For synthetic data, we normalize each component to have the same mean value (over the features) as the matching ground truth component. For real data, we use a healthy reference — a vector that represent the typical signal of healthy samples (obtained from (Sadeh et al., 2021)). We choose a group of house-keeping genes, that are known to have high signal and low variation between cell-types and tissues (Sadeh et al.). We then normalize each component to have the same median value these house-keeping genes as the healthy reference. This normalization procedure is meant to anchor all components' house-keeping genes (that have similar behavior in different cell-types) to a similar location.

Other matrices for which Eq. 42 holds can be used to scale only NMF and not VarNMF solutions, as it will cause different components to become correlated, thus the scaled solution will be distinct from the original solution (in which all features are independent). These matrices can sometimes exists, and the conditions for essential uniqueness of the NMF solutions are complicated (Donoho & Stodden, 2003). Fortunately, in practice, following some basic sparseness properties, it is likely that NMF will have an essentially unique solution (Huang et al., 2013). In our case, these conditions usually hold, and we will assume the NMF solution (normalized for permutations and normalization matrices) is unique.

### A.7 GENERALIZATION

The train log-likelihood measure evaluates the ability of an algorithm to fit a dataset, and is a popular measure in the world of probabilistic modeling and unsupervised learning (which includes NMF research). However, in our case and in many other situations, the overall goal is extract an accurate distribution of features (e.g. genes). While the mixing weights are specific to each sample, the learned distributions are general and should fit all samples, including unseen ones. Therefore, we want measure the generalization abilities of our model on a new dataset sampled using the same distribution.

To test the generalization abilities of VarNMF, we use the learned distributions to learn a new $W^{\text{test}}$ for a test dataset $V^{\text{test}}$ by applying EMcAB (Appendix A.5). Similarly, for NMF, we use the learned constant components $H$ to learn $W^{\text{test}}$ by applying the NMF Multiplicative update rule (Lee & Seung, 2000) on $V^{\text{test}}$ and keeping $H$ constant. In both cases we report the test log-likelihood to be the log-likelihood for the test data with the train distributions or components and the test mixing weights.

## B SYNTHETIC DATA

### B.1 RESULTS FOR DIFFERENT K VALUES

We examine the results NMF and VarNMF on synthetic datasets with increasing source variation and increasing number of sources (Section 4.2 for more details). In Fig. 6A we observe a decline in NMF train and test log-likelihood results as the levels of variation in the dataset (controlled by the coefficient of variation of the ground truth source distributions) increase. This is true for all values of $K$ (#Components in the dataset) but is most apparent for small values of $K$. We conclude that for that VarNMF better captures the datasets distribution in the presence of high component variation.

Next, we examine the learned parameters of each model against the ground truth. Starting with the learned weights of NMF and VarNMF against the ground truth (Fig.. 6B), the two models have almost identical values, with performances decreasing with the level of variation. As for learned components against the ground truth, the VarNMF component means are closer to the ground truth

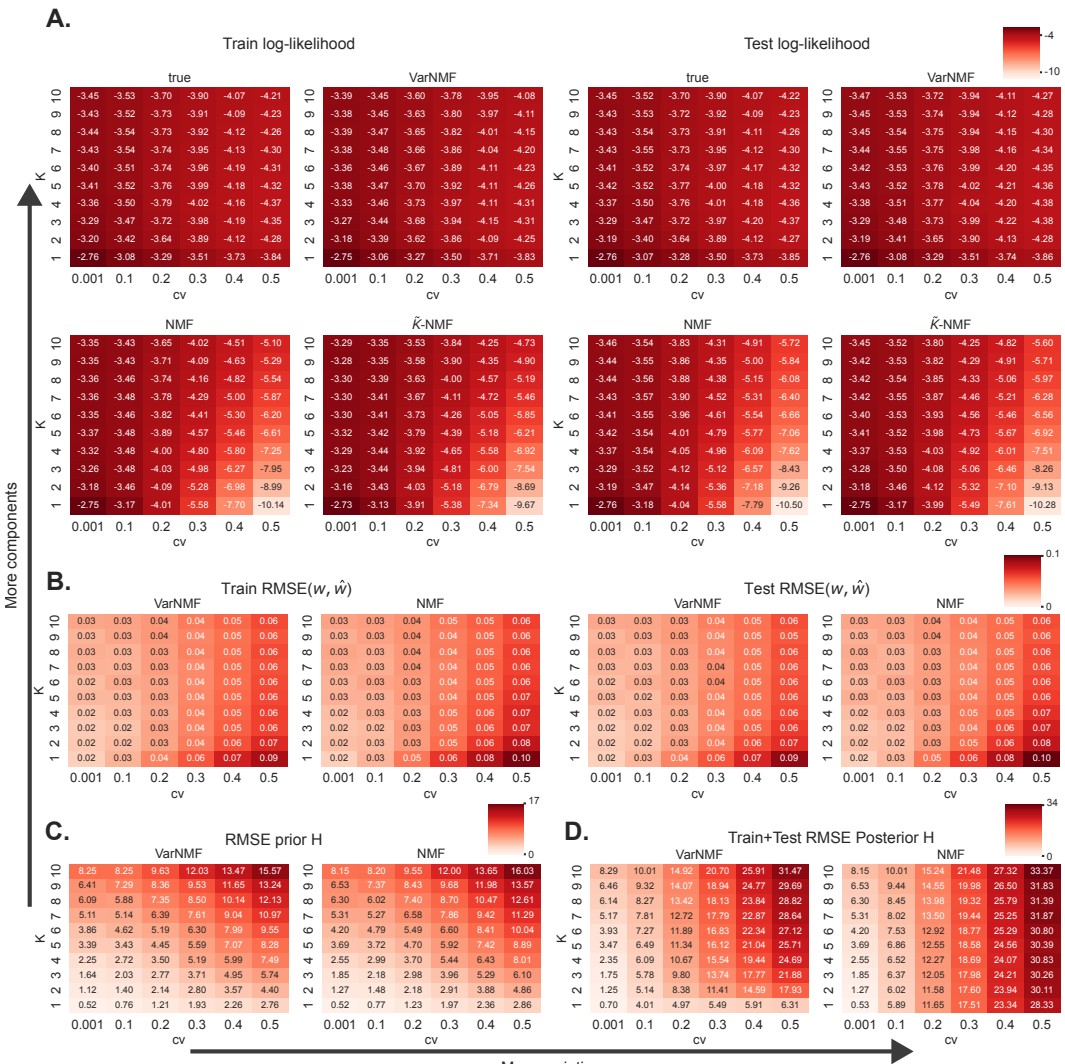

Figure 6: Decomposing synthetic data with $K = 1, ..., 10$ components: A) Train and test log-likelihood of the ground truth parameters and three models - NMF, VarNMF and $\tilde{K}$-NMF (NMF with higher degrees of freedom than VarNMF), versus the coefficient of variation of the dataset and versus $K$. The log-likelihood values are normalized to nats/observation. B) Root mean square error (RMSE) of the the train and test weights estimated by VarNMF and NMF versus the ground truth weights. C) RMSE of the mean of the distributions estimated by VarNMF and the constant components estimated by NMF versus the mean of the ground truth distributions. D) RMSE the per-sample posterior expected signal estimated by VarNMF and the constant components estimated by NMF versus the ground truth $H[i]$. The shown values are the mean over $T = 10$ runs.

means than NMF's estimates (Fig.. 6C) but generally the trends look similar: The RMSE values increase with the levels of variation and with the number of components. This suggests that component variation between samples increases the complexity of the data and hinder both algorithms effort to extract the ground truth weights and mean of the sources distributions. However, comparing the ground truth per-sample contribution of source $k$, $H[i]_k$, to the VarNMF posterior expected signal versus NMF constant components (Fig. 6D), the VarNMF RMSE is similar to the NMF values for datasets with no variation (cv= 0.001), but outperform the NMF solution when this variation increases, and NMF perform poorly for every value of $K$.

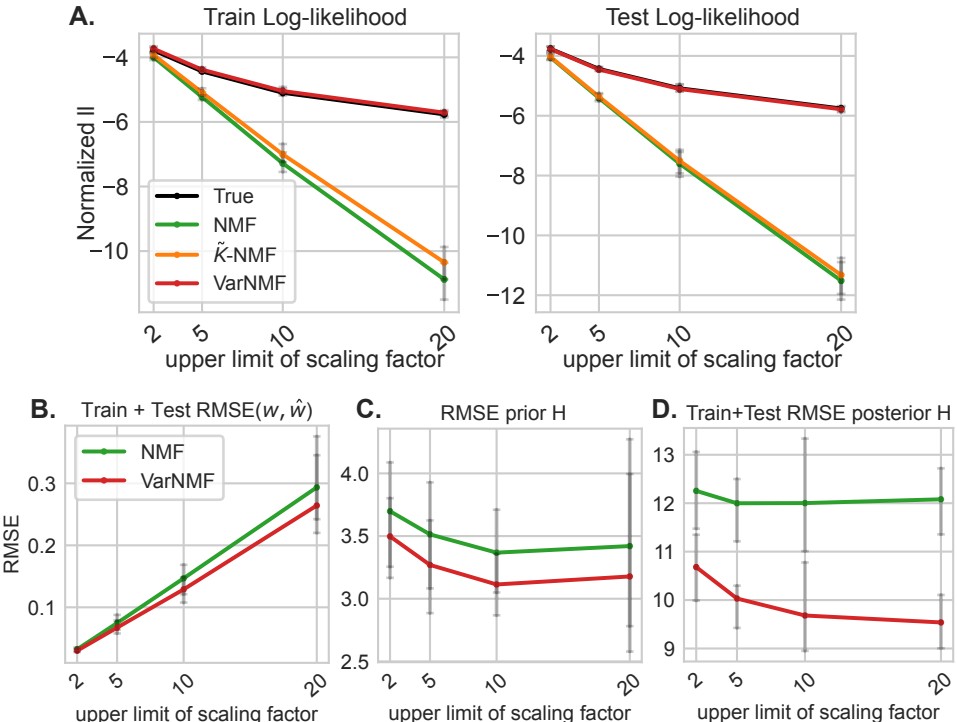

Figure 7: Decomposing synthetic data with increasing sampling noise: A) Train and test log-likelihood of the ground truth parameters and three models - NMF, VarNMF and $\tilde{K}$-NMF, versus the upper limit of the scaling factors distribution. The log-likelihood values are normalized to nats/observation. B) RMSE of the the train and test weights estimated by VarNMF and NMF versus the ground truth weights. C) RMSE of the mean of the distributions estimated by VarNMF and the constant components estimated by NMF versus the mean of the ground truth distributions. D) RMSE the per-sample posterior expected signal estimated by VarNMF and the constant components estimated by NMF versus the ground truth $H[i]$. The shown values are the mean over $T = 10$ runs of datasets with $K = 4$ components and coefficient of variation cv $= 0.2$

## B.2 ROBUSTNESS TO SAMPLING NOISE AND NUMBER OF SAMPLES

In Fig 7 we examine the effect of increasing sampling noise on the performances of NMF and VarNMF. This is done by increasing the limits of the uniform distribution from which we sample the per-sample scaling factors $\lambda[i]$ (Section 4.2 for more details). Specifically, applying VarNMF and NMF to datasets with scaling factors sampled from $\mathcal{U}\left(\left[\frac{l}{2}, l\right]\right)$ for $l = 2, 5, 10, 20$, results in an expected drop in performances. However, the NMF performances drop significantly more than those of VarNMF, suggesting that the advantage of VarNMF is robust to sampling noise.

Additionally, when increasing the number of samples $N$, we get better estimations of the ground truth parameters, but also a consistent advantage of VarNMF over NMF (Fig 8).

## C CFCHIP DATA

We collected a dataset of cfChIP-seq samples from Sadeh et al. (2021) and Fialkoff et al. (2022). This data includes plasma samples of 80 healthy subjects, 139 small-cell lung cancer (SCLC) patients, and 86 colorectal cancer (CRC) patients. This data has two representations.

- Normalized reads - gene values after normalization, According to Sadeh et al. (2021) the normalization ensure that the samples agree on a set of reference genes.

- Raw counts - gene values are the observed count in each sample.

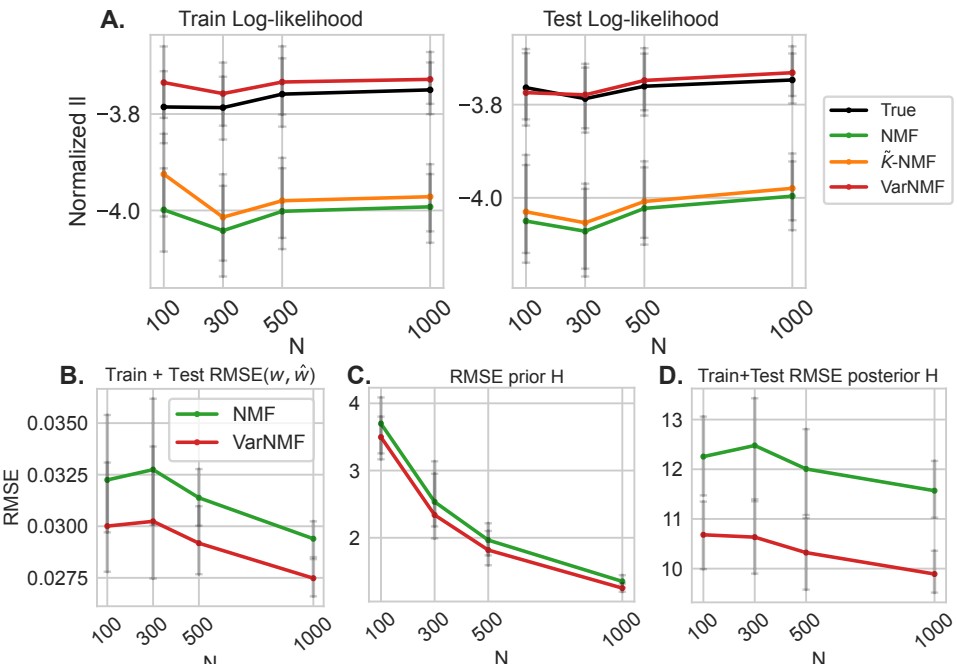

Figure 8: Decomposing synthetic data with increasing number of samples $N$: A) Train and test log-likelihood of the ground truth parameters and three models - NMF, VarNMF and $\tilde{K}$-NMF, versus the number of samples $N$. The log-likelihood values are normalized to nats/observation. B) RMSE of the the train and test weights estimated by VarNMF and NMF versus the ground truth weights. C) RMSE of the mean of the distributions estimated by VarNMF and the constant components estimated by NMF versus the mean of the ground truth distributions. D) RMSE the per-sample posterior expected signal estimated by VarNMF and the constant components estimated by NMF versus the ground truth $H[i]$. The shown values are the mean over $T = 10$ runs of datasets with $K = 4$ components and coefficient of variation cv $= 0.2$

We used the normalized reads representation for selecting genes (below) and for PCA analysis (Fig. 4). For the actual data processing we used raw counts, and let the model fit scaling parameter per sample.

### C.1 CHOOSING DATASET FEATURES

To choose relevant features we first excluded genes that are positioned on chromosomes X or Y (sex-specific genes) and putative genes (ORFs without a name, pseudogenes and such).

For each gene we computed several statistics across the entire dataset — mean $\mu_g$, variance $\sigma_g^2$, coefficient of variation $\eta_g = \sigma_g/\mu_g$, number of times they are above 0 $n_g$, and maximal value $m_g$. Based on these we defined three groups of genes:

- Housekeeping-like genes [6029 genes] – Genes with high levels ($\mu_g > 50$) and low variability ($\eta_g < 0.5$)

- Variable genes [5119 genes] – genes that do not appear in the first group and have high variation ($\eta_g > 0.25$), expression in more than 50 samples ($n_g > 50$), and some observations above a threshold ($m_g > 10$).

- Remaining non-excluded genes [5764 genes]

We reasoned that the housekeeping-like genes provide stability and anchor the estimation of values. The variable genes provide a chance to identify interesting phenomena. Thus we randomly selected 5000 variable genes, and added 1000 randomly selected genes from each of the two other groups.

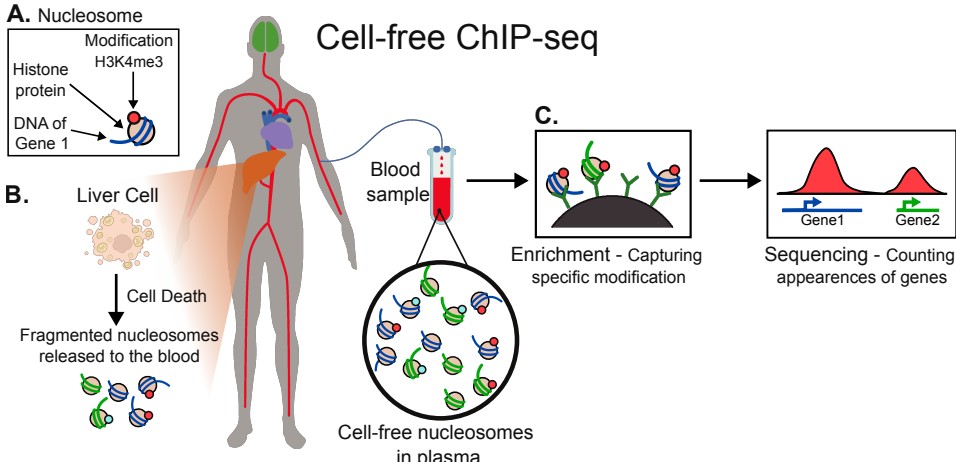

Figure 9: cfChIP concept - A) Genomic DNA is present in the nucleus of every human cell, and is packaged into nucleosome complexes made up of DNA wrapped around histone proteins, which can be modified in a way that is tightly coordinated with gene activity (Soares et al., 2017). B) Upon cell-death, the genome is fragmented and nucleosomes are released into the circulation as cell-free nucleosomes that retain their modifications. C) Recent method (Sadeh et al., 2021) allows capturing the modified nucleosomes from plasma, and then sequence the DNA fragments bound to these nucleosomes. Mapping these sequences to the genome, we can associate them with genes. Thus, similar to RNA-seq data, this assay provides a quantative signal of activity associated with each gene. This signal reflects the aggregate contribution of all cells that released modified nucleosomes into the circulation, and thus if we could break it into individual components, we would be able to report on each sub-population of cells (e.g., tumor cells, immune cells).

Table 1: Enrichr results for NMF solution, $K = 4$

| Component | Database | Term | Overlap | Adj. p-value |
|---|---|---|---|---|
| 1 37 genes | BioPlanet 2019 BioPlanet 2019 | Hemostasis pathway Platelet activation signaling and aggregation | 12/468 8/205 | 3.52e-09 2.19e-07 |
| 2 297 genes | ARCHS4 Tissues ARCHS4 Tissues ARCHS4 Tissues | PERIPHERAL BLOOD NEUTROPHIL MACROPHAGE | 165/2316 140/2316 137/2316 | 8.93e-75 9.49e-52 1.88e-48 |
| 3 464 genes | Cancer Cell Line Enc. Cancer Cell Line Enc. Cancer Cell Line Enc. | CORL279 LUNG CORL24 LUNG NCIH446 LUNG | 116/465 117/569 94/327 | 1.10e-84 1.98e-75 3.20e-74 |
| 4 329 genes | Cancer Cell Line Enc. Cancer Cell Line Enc. Cancer Cell Line Enc. | SNU283 LARGE INTESTINE CL40 LARGE INTESTINEE SW1463 LARGE INTESTINE | 58/189 55/210 49/177 | 5.99e-55 5.63e-48 6.22e-44 |

## C.2 TESTING GENE-LISTS VERSUS CURATED DATABASES

To choose genes that differ in one component $k$ from the rest, we first discard genes that have a value lower than 10 in the specific component (or in the mean of the component, in the case of VarNMF). We then calculate the median value across all components for each gene, and sort the genes in descending order based on the fold-change increase observed in component $k$ relative to the calculated median. Genes with less than a 2-fold increase are excluded.

Table 2: Enrichr results for VarNMF solution, $K = 4$

| Component | Database | Term | Overlap | Adj. p-value |
|---|---|---|---|---|
| 1
28 genes | Reactome 2022
BioPlanet 2019
BioPlanet 2019 | Hemostasis
Hemostasis pathway
Platelet Activation,
Signaling and Aggregation | 10/576
11/468
8/205 | 3.11e-07
2.12e-09
1.91e-08 |
| 2
118 genes | ARCHS4 Tissues
ARCHS4 Tissues
ARCHS4 Tissues
ARCHS4 Tissues | PERIPHERAL BLOOD
NEUTROPHIL
GRANULOCYTE
MACROPHAGE | 86/2316
76/2316
71/2316
66/2316 | 1.24e-52
1.45e-40
3.56e-35
4.05e-30 |
| 3
700 genes | Cancer Cell Line Enc.
Cancer Cell Line Enc.
Cancer Cell Line Enc. | CORL279 LUNG
NCIH446 LUNG
CORL24 LUNG | 155/465
122/327
150/569 | 8.09e-108
1.50e-90
6.80e-88 |
| 4
383 genes | Cancer Cell Line Enc.
Cancer Cell Line Enc.
Cancer Cell Line Enc.
Cancer Cell Line Enc. | JHH5 LIVER
C3A LIVER
SNU283 LARGE INTESTINE
SW1463 LARGE INTESTINE | 89/245
95/387
53/189
51/177 | 5.58e-88
3.12e-76
9.57e-45
1.03e-43 |

Next, we test whether these genes are significantly over-represented in curated genes-lists associated with a specific tissue or cell-type (using the Enrichr tool, Chen et al. (2013) for details). The top results for each component of the NMF and VarNMF solutions with $K = 4$ components are presented in Tables 1, 2. For the "healthy" components we used reference data from human tissues and cell-types (*Reactome 2022*, *BioPlanet 2019* and *ARCHS4 Tissues*). For the disease-associated components, we made use of the *Cancer Cell-Line Encyclopedia* (CCLE) which contains data from cell of various cancers.

Results for the first three components are similar between the two models. The first two components' genes are strongly enriched for Platelets and Neutrophils, which are the two main sources of cell-free DNA in healthy samples (Sadeh et al., 2021). The second component is also enriched for Macrophages that differentiate from Monocytes which are also found in high concentrations in cell-free DNA from healthy samples. The third component (with non zeros weights mostly in SCLC patients) is enriched specifically for SCLC-derived cell-lines in both the NMF and VarNMF solutions. This indicates that this component indeed represents the tumor derived cell-free DNA in the SCLC patients plasma. The forth component (with non zeros weights mostly in CRC patients) displays different associations between the two models: The NMF solution is enriched exclusively for colon-derived cell-lines, aligning with the CRC patients diagnosis. On the other hand, the VarNMF solution is enriched for both liver and colon derived cell-lines. The former enrichment may reflect liver metastasis which exist in many of the CRC patients in this cohort.

## C.3 RESULTS OF NMF AND VARNMF FOR A RANGE OF KS

We examine decomposition of the NMF and VarNMF models and algorithm on cfChiP-seq dataset with increasing number of components $K = 2, ..., 6$ (Section 4.3 for more details). In Fig. 10 and 11 we look at the resulting weights. We observe similar results in the train and test datasets as well as for both NMF and VarNMF. However, in the NMF solution we observe a single healthy component (that is high in healthy samples but also appear in a range of weights in the cancer samples) whereas VarNMF displays two. Additionally, starting from $K = 3$, there are at least two components that are cancer-specific (that have non-zero weights mostly in one cancer type). The one SCLC-associated component of $K = 4$ splits into two SCLC-associated components starting from $K = 5$, and the CRC-associated component remains unique in all solutions.

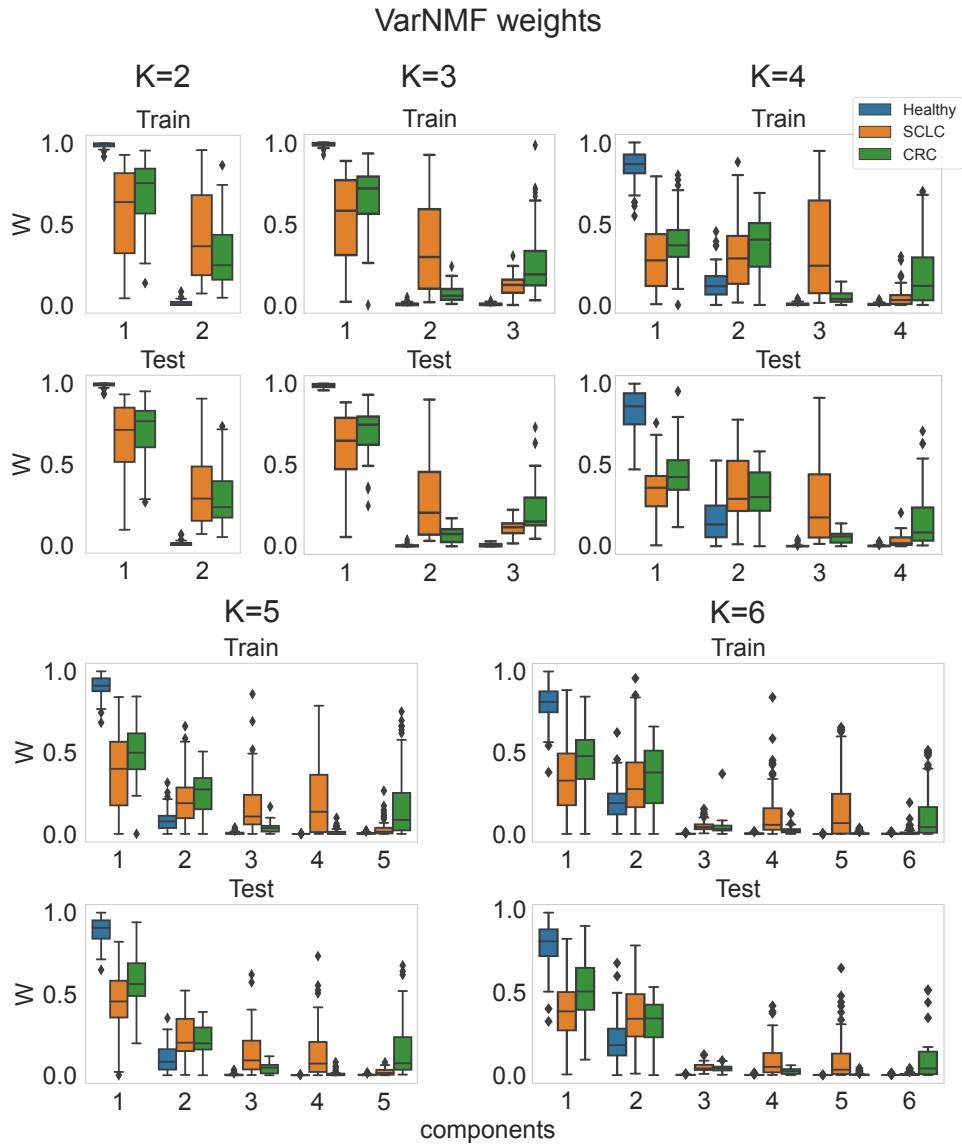

Figure 10: Decomposing cell-free ChIP-seq data with VarNMF: The VarNMF estimated train and test weights for each component in the $K = 2, ..., 6$ solutions, aggregated by sample sub-cohort (healthy, SCLC and CRC). Weights are normalized so that each sample has a total weight of $1$.

Lastly, we observe a shared-cancer component in the $K = 4, 5$ solutions. In VarNMF it is also shared with healthy samples and is associated with genes of blood-related cell-types, which is expected (Appendix C.2). In NMF, however, this component is not shared with healthy samples, but is still associated with these cell-types. Starting from $K = 6$, the two models results in two healthy components, and one shared cancer component. This suggests that both models extract a component that represent shared features between the cancers.

We conclude that both VarNMF and NMF result in similar weights, with one or two healthy components that reflect the healthy cell-free material in the plasma, one or two cancer-specific components for each type of cancer, that represent the disease contribution to the plasma, and possibly one component that is shared between the cancers.

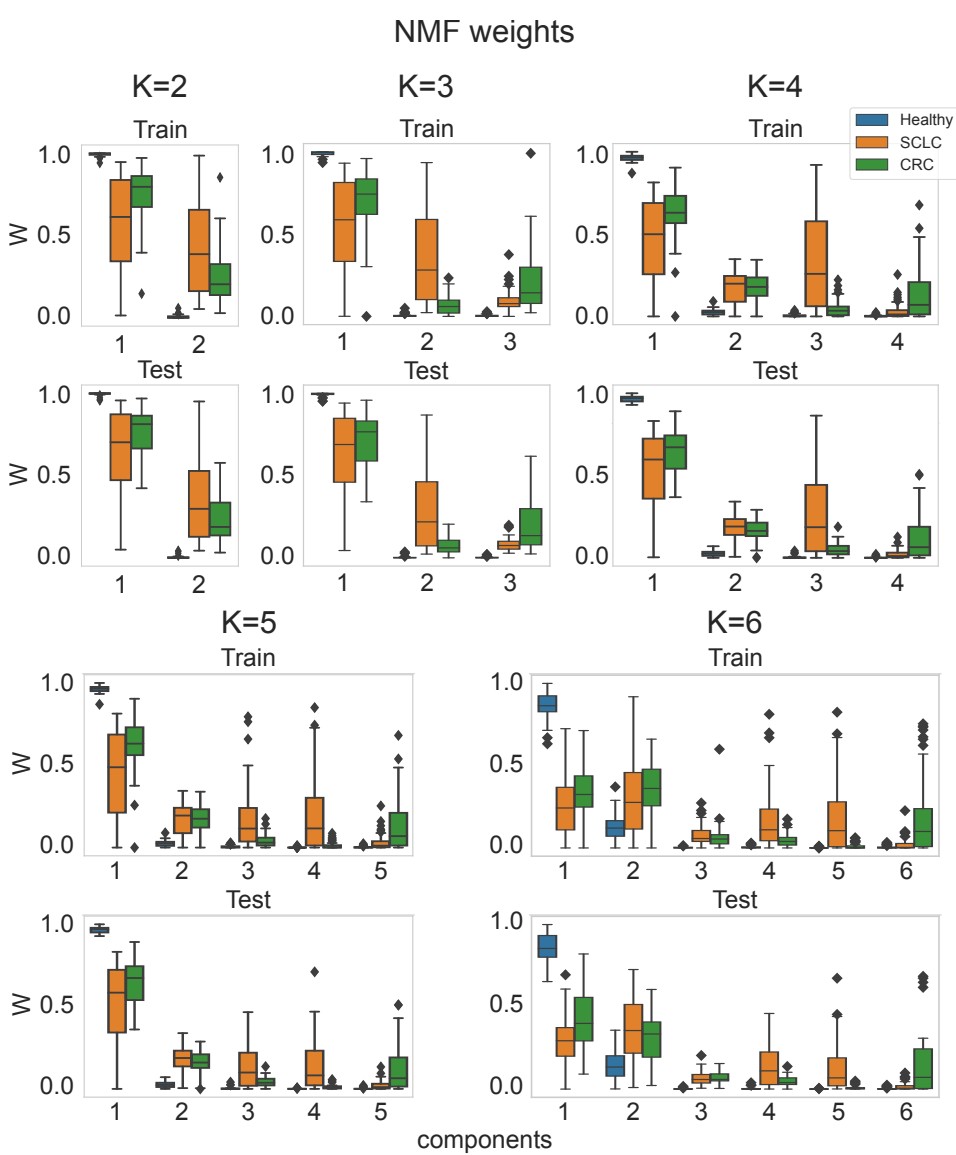

Figure 11: Decomposing cell-free ChIP-seq data with NMF: The NMF estimated train and test weights for each component in the $K = 2, ..., 6$ solutions, aggregated by sample sub-cohort (healthy, SCLC and CRC). Weights are normalized so that each sample has a total weight of 1.

