# OpenReview forum: "Non-negative Probabilistic Factorization"
_ICLR.cc/2024/Conference — Submitted to ICLR 2024_

### Official Review · Reviewer_6VBV · 2023-10-17

**Soundness:** 2 fair
**Presentation:** 2 fair
**Contribution:** 1 poor
**Rating:** 3
**Confidence:** 4

**Summary:**

In this submission, the authors claim to propose a probabilistic extension of NMF, namely VarNMF, that introduces a novel type of variation between samples. They learn this variation through a straightforward expectation-maximization procedure. The proposed method is demonstrated on a dataset of genomic measurements from liquid biopsies to show its effectiveness.

**Strengths:**

The performed experiments on real data seem to be of some interest.

**Weaknesses:**

- Equations (2), (3), (4) and Figure 1 are fairly standard for people familiar with NMF. However, the authors dedicate considerable space to these concepts and even replicate them later, such as in Equations (8) and (9).

- It appears that the proposed VarNMF is a simplified version of the model/framework designed in

Tan, V. Y., & Févotte, C. (2012). Automatic relevance determination in nonnegative matrix factorization with the/spl beta/-divergence. IEEE transactions on pattern analysis and machine intelligence, 35(7), 1592-1605.

The authors of the current submission should not be unaware of this closely related work.

- The mathematical notations in the paper are somewhat messy. For example, the dimension of $V$ is first mentioned on page 3, but $H_k \in \mathbb{R}^M_{\ge 0}$ suddenly appears on page 2. On page 4, the notation $V = (V[1], \ldots, V[N]) \in \mathbb{R}^{N\times M}_{\ge 0}$ is written in a confusing style. Additionally, some simple equations such as (8), (9), and (12) are repeated (although notations are slightly different) and occupy entire lines.

**Questions:**

No

---

> ### Author Response · Authors · 2023-11-13
> **Response to Reviewer 6VBV**
>
> **[Novelty]** &emsp; We politely disagree with the reviewer and believe that while equation (2) describes the standard formulation of NMF, Eqs. 3 and 4 provide a new model that includes a per-sample latent component matrix H[i] sampled from learned prior distributions - unlike previous probabilistic\bayesian extensions of NMF, where the prior distributions are assumed to be pre-learned and the component matrix is shared between all samples. We will add to the paper a more detailed discussion on this type of related work to highlight the inherent differences.
>
> Moreover, Eqs. 8 and 9 reformalize Eqs. 3 and 4 using auxiliary variables to provide a more efficient way of computing the model’s likelihood.
>
> **[Suggested literature]** &emsp; We thank the reviewer for suggesting this interesting paper. However, we politely disagree with the statement regarding the relationship between the two approaches. The main point of [TF12] is an approach for model selection in NMF (defined with generalized loss function), and specifically for the selection of the number of sources K that contribute to the mixed signal. While this is an important subject in NMF literature, it is orthogonal to the extension discussed in our work.
>
> **[Notation]** &emsp; We are sorry that the notation is not clear. We are adding a notation section to provide a clear reference for the notation we use. We hope that this will alleviate the confusion.

---

> > ### Comment · Reviewer_6VBV · 2023-11-14
> > **Response**
> >
> > Thanks for the response.
> >
> > If the authors spend a little more time on the work [TF12], they will find that the algorithms (see Algorithms 1 and 2) proposed therein can also estimate the component matrices as a by-product.
> >
> > I still think that the problem studied in this work is a simplified version of those in [TF12], although using a slightly different EM-type approach ([TF12] also proposes EM/MU (Multiplicative Updates)-type algorithms).
> >
> > As Reviewer oPxL also pointed out, probabilistic/Bayesian NMF has been studied extensively, with various EM/MU algorithms. There is a significant lack of discussion about relevant prior works and comparative experiments with related approaches.
> >
> > Therefore, I would like to keep my original score.

---

> ### Author Response · Authors · 2023-11-15
> **Response to Reviewer 6VBV (1/2)**
>
> We would like to stress again the fundamental difference between our work and the different probabilistic/Bayesian versions of NMF, including the one suggested in [TF12].
>
> We emphasize that the novelty is *not* in the algorithm (which is a fairly standard graduate-level exercise EM in our case) but in the probabilistic model itself. Thus while our learning and inference algorithms are different from those of previous works, it is due to the differences in the mathematical models. Therefore we will not get into the optimization algorithms employed by the different papers.
>
> To highlight the differences in the *mathematical models*, we attach below a description of the model in our work versus the standard NMF [1], Bayesian NMF [2], and [TF12] suggested by the reviewer. We hope this will clarify any misunderstanding. Additionally, as mentioned, we will add to the paper a more detailed discussion on this type of related work to highlight the inherent differences.
>
> First, we note that [1] and [TF12] refer to W as the basis\component KxM matrix and to H as the activation\weights NxK matrix (where N=#samples, M=#features and K=#components). As the component and weight matrices are symmetrical in [1, TF12] but not in our work, we will switch the names of W,H when describing [1, TF12]. In [2], the notations are completely different and we will match them to our notation for an easier read.
>
> **The plate notations of the different models are provided in https://ibb.co/J3zWh85.**
>
> These translate into the following generative models, where
>
> 1. For each random variable $X$, $p_X(\theta^X)$ is the distribution of $X$ in the relevant model with some parameters $\theta^X$.
> 2. We index samples\ data-points with $i=1,…,N$, features with $j=1,…,M$, and components with $k=1,…,K$.
> 3. A random variable $X$ is indexed by sample $i$ using square parentheses $X[i]$, by $j$ and $k$ using subscript $X_{k}$\ $X_{j}$\ $X_{k,j}$, and by both as $X[i]\_{k}$\ $X[i]\_{j}$\ $X[i]\_{k,j}$ (as mentioned, a more explicit notation description will be added to the paper).
>
> **NMF:** $W,H$ are parameters
>
> $\forall\_{i,j}, V[i]\_{j}\sim p_V(\sum_k W[i]\_{k} \cdot H\_{k,j})$
>
> **Bayesian-NMF:** $\theta^W, \theta^H$ are hyperparameters
>
> $\forall\_{i,k}, W[i]\_{k} \sim p\_W(\theta^W\_{i,k})$
>
> $\forall\_{k,j}, H\_{k,j} \sim p_H(\theta^H\_{k,j})$
>
>  $\forall\_{i,j}, V[i]\_{j}\sim p_V(\sum_k W[i]\_{k} \cdot H\_{k,j})$
>
> **[TN12]:** $a,b$ are hyperparameters
>
> $\forall\_{k}, \lambda\_{k} \sim p\_\lambda(a,b)$
>
> $\forall\_{i,k}, W[i]_{k} \sim p\_W(\lambda\_{k})$
>
> $\forall\_{k,j}, H\_{k,j} \sim p\_H(\lambda\_{k})$
>
>  $\forall\_{i,j}, V[i]\_{j}\sim p\_V(\sum_k W[i]\_{k} \cdot H\_{k,j})$
>
> **VarNMF:** $W,\theta_H$ are parameters
>
> $\forall\_{i,k,j}, H[i]\_{k,j} \sim p_H(\theta^H\_{k,j})$
>
>  $\forall\_{i,j}, V[i]\_{j}\sim p_V(\sum_k W[i]\_{k} \cdot H[i]\_{k,j})$
>
> Note the difference in the indexing of $H$ in VarNMF compared to all other models - it is now indexed by ******sample****** $i$ and not only by component $k$ and feature $j$. In the plate models this is reflected by the fact that in VarNMF, the red (sample) plate includes $H$. This is not the case in all other models.
>
> This difference is the main point of our work - we have a latent component matrix $H[i]$ **for each sample $i$**, in contrast to a single (latent or not) component matrix $H$ in all other models. This change is also reflected in the sampling of $V$, which is identical for all previous models (all samples share the matrix $H$), but different in ours (each sample $i$ is sampled using a different matrix $H[i]$).
>
> The relevance of defining a component matrix $H[i]$ for each sample $i$ is that now we can ask about the distribution of these matrices in the population - i.e., about the distribution $p_H$ that is defined over samples (specifically - estimate its parameters $\theta^H$ from data). This is not done in the other models mentioned above - either because there is no distribution on $H$ (NMF), or because the distribution  $p_H$ reflects prior knowledge\ regularization purposes (e.g., non-negativity\ sparsity constraints in Bayesian-NMF, cost on higher dimensionality in [TF12]) and not variance between samples\ data-points.
>
> Specifically, the prior distributions $p(W[i]\_{k}|\lambda\_k)$,  $p(H\_{k,j}|\lambda\_k)$ in [TF12] makes $W[i]\_k$  and $H\_{k,j}$ to be “[…] tied together by a common variance-like parameter $\lambda\_k$ […]. When a particular $\lambda\_k$ is small, that particular column of W and row of H are not relevant and vice versa.” [TF12] - i.e., the distribution $p\_H$ does not reflect variance in the values of $H$ between samples. This is simply a prior distribution that drives the values of $W,H$ down to regularize the number of components $K$.
>
> With this explanation, we feel like we demonstrated the fundamental differences between our work and previous works, including [TF12]. We hope this clarifies any misunderstanding.

---

> > ### Comment · Reviewer_6VBV · 2023-11-22
> > **Further response**
> >
> > I appreciate the authors' effort in providing a detailed response.
> >
> > However, I agree with Reviewer oPxL that the problem remains that your paper lacks a proper discussion about many relevant and related prior works and it also does not include convincing experimental evaluations in which you empirical compare VarNMF to related NMF approaches. I believe that including detailed discussions to closely relevant prior works and extensive experimental comparisons with related methods would be helpful in the resubmitted version.

---

> ### Author Response · Authors · 2023-11-15
> **Response to Reviewer 6VBV (2/2)**
>
> References:
>
> [1] Lee, D., & Seung, H. S. (2000). Algorithms for non-negative matrix factorization.
>
> [2] Schmidt, M. N., Winther, O., & Hansen, L. K. (2009). Bayesian non-negative matrix factorization.
>
> [TF12] Tan, V. Y., & Févotte, C. (2012). Automatic relevance determination in nonnegative matrix factorization with the/spl beta/-divergence.

---

> > ### Comment · Reviewer_oPxL · 2023-11-20
> >
> > Although you included a discussion as to why you deem your approach novel,
> > the problem remains that your paper lacks a proper discussion about many relevant and related prior works and it also does not include convincing experimental evaluations in which you empirical compare VarNMF to related NMF approaches.
> >
> > I agree with reviewer 6VBV and do not plan to raise my score.
> > This work requires a major revision to address all open problems.
> > I do believe that a resubmission will be fruitful, as VarNMF might bare some potential.

---

### Official Review · Reviewer_UHns · 2023-10-25

**Soundness:** 3 good
**Presentation:** 4 excellent
**Contribution:** 3 good
**Rating:** 8
**Confidence:** 3

**Summary:**

Non Negative Matrix Factorization can be understood as a particular case of blind source separation. In blind source separation problems, we are given several measurements (the so-called channels) corresponding to noisy observations of the mixing of different sources. The goal is to recover the sources (and the mixing components of the different sources for each channel). One main limitation of NMF models is that the sources are considered to be the same across the different channels: an assumption that is not always satisfied in practice.

In this work, the authors propose an extension of the standard NMF model where the different source signals are sampled from some probability distribution in the feature domain for each channel. Contrary to previous works, the distribution of each source is not pre-learned assuming that observations of the sources are available but they are inferred from the channel data.

**Strengths:**

- The paper is well written and provide a clear description of previous works in the field of non negative matrix factorization.

- The authors conduct experiments on both simulated and real world datasets. They show that their method leads to better results to take into account the variability in the sources.

- The authors properly address several important aspects of their method. In particular, they provide in the Appendix a detailed explanation of the way to mitigate the heavy computational cost of the E-step of their EM procedure when dealing with real data and a large amount of features.

**Weaknesses:**

- One possible limitation of the proposed approach is the assumption that the probability distribution of source $k$ can be factorized in the feature domain (i.e. independence of the features of source $k$ is considered). This assumption might be limiting the expressivity of the proposed model. In particular, it would have been particularly interesting to go beyond this independent assumption for the application on real data since it is well-known that genes can be up or down regulated by other genes.

- Another limitation of the method is the heavy computational cost of the algorithm, even if the authors propose approaches to cope with this issue.

- The method proposed by the authors is specific to the Poisson distribution for the observational probability (and to the assumptions made on the dependence structure of the sources).

**Questions:**

I thank the authors for this very nice work. The paper is well written and it was a pleasure to read it. My question are the following.

- How the method can be adapted to consider other observational probability such as a negative
binomial distribution (instead of the Poisson) ? The negative binomial distribution would be particularly valuable for the biological application suggested by the authors as it enables the modeling of overdispersion, a phenomenon known to occur in such biological settings.

- In the same spirit, would it be possible to change the model to have dependence between features of source $k$ ?


Here are some typos (or minor comments):

- It might be beneficial to also include a reference to Section A.7 of the Appendix in the section discussing experiments on real data, which explains how you tested your model on fresh data. Indeed, I was briefly confused because NMFs are known to encounter the cold start problem, and I was curious about the authors' approach to testing the model on new data.

- At Eq.(11), a closing parenthesis should be removed. Same at Eq.(35).

- At page 11, "Differentiating" should be "differentiating".

- In Lemma 5, it might be good to specify which parametrization of the negative binomial distribution you consider.

- I think the logarithm should be removed in Eq.(21).

- In Eq.(30), the symbol "sum" does not display properly.

- After Eq.(37), the same style is not used for the log-likelihood function ($l$ instead of $\ell$).

- At Eq.(40) (second line), I think you should remove one "$p(.)$".

---

> ### Author Response · Authors · 2023-11-13
> **Response to Reviewer UHns**
>
> We thank the reviewer for the kind words and extensive recommendations. We will fix all typos.
>
> **[Independence of features]** &emsp; We definitely agree with the reviewer’s comment regarding the limitation of the independence assumption (see our last paragraph) and are actively exploring different models for capturing dependencies between genes (ranging from network of dependencies to lower-dimensional latent representations). This also relates to your second question.
>
> **[Computational costs]** &emsp; Again, we agree with the reviewer's comment on the computational costs and we are exploring different computational approaches (e.g., variational inference) to reduce the computational costs. We decided to focus here on the basic idea, as the paper is already dense.
>
> **[Observational probability]** &emsp; Indeed, the model is specific to Poisson distribution. There are many applications where the Poisson distribution is the natural model of technical variability, including most sequencing-based assays. More complex distributions used in the literature, such as Negative Binomial distributions essentially fold the Poisson noise with additional levels of uncertainty that we claim are often better modeled as source variation (wrt to your first question). With that said, it is of interest to explore the feasibility of other noise models. A move to variational inference, for example, will provide flexibility with respect to the observational distribution.

---

> > ### Comment · Reviewer_UHns · 2023-11-22
> >
> > I thank the authors for their response to my initial review of the manuscript. I appreciate the time and effort invested in addressing the concerns raised during the review process.
> > Based on the reviews and discussions with the other reviewers, I realized that some important connections with prior work should be better discussed in the manuscript. In my opinion, the paper is well written and was interesting to read but I was not aware of some previous works that seem to be very close to the method presented by the authors (cf. discussion with reviewer 6VBV). I have decided to keep my current rating of the manuscript but I decreased my confidence score.

---

### Official Review · Reviewer_oPxL · 2023-10-31

**Soundness:** 3 good
**Presentation:** 3 good
**Contribution:** 2 fair
**Rating:** 3
**Confidence:** 4

**Summary:**

The paper introduces a probabilistic non-negative matrix factorization (VarNMF) in which components vary per sample, rather than being "static" coefficients.
To achieve this variability, the authors introduce a Poisson Factorization formulation in which per-observation-components are independently distributed.
The result is an intractable likelihood function that the authors simplify in terms of additional auxiliary variables, which enables model-estimation using EM.
In two case-studies, the authors apply VarNMF to bioinformatics problems.

**Strengths:**

The per-sample variability increases the modeling space.
The illustration (Fig 1) is insightful (but could be significantly smaller).
The Bayesian factorization model and resulting likelihood has been described well.
The bioinformatics experiments are detailed.
The bioinformatics data has been described in detail (potential candidate for the appendix)

**Weaknesses:**

1. The related work has not been discussed thoroughly.
The paper only shallowly discusses related work on Poisson Factorization and misses out on Poisson Factor Analysis.
Probabilistic Matrix Factorization and Bayesian Matrix Factorization are not discussed in detail.

2. Given that the authors consider the 'mean signals' of the variable components in their interpretation and analysis, it is not clear if one could achieve the same by using a Bayesian Matrix Factorization and an appropriate prior distribution.
Given the lack of contextualization, it is not clear how novel the proposition is.

3. The paper lacks in experimental contextualization.
That is, the paper does not properly compare VarNMF to the state-of-the-art of direct competitors from Poisson Factorization, Poisson Factor Analysis, or Bayesian Matrix Factorization. Comparing with Hierarchical Poisson Factorization (HPA) as a baseline might be interesting.
That means, benefits and limitations of VarNMF compared to the state-of-the-art remains unclear.

4. The experimental section lack in breadth.
Since the focus lies on bioinformatics case-studies, it is unclear how VarNMF performs in other domains.
Considering the interests of our ICLR community, readers probably expect a broader evaluation.
Given the related work, it would be highly beneficial to include experiments on recommender systems or computer vision datasets, such as Movielens, Netflix Prize (which all are ideally suitable for the Poisson Factorization), or Olivetti Faces.

5. The empirical evaluation lacks in depth.
The robustness of VarNMF under noise has not been evaluated.
An empirical evaluation on convergence is not included.
The paper does not compare the runtime of VarNMF to competing methods.
The performance characteristics of VarNMF on a wide range of datasets is unclear.

6. The paper does not include a Reproducibility Statement and the submission does not include a Reproducibility Package.

7. The title of this paper is quite general and does not match the specific contribution of this paper.

**Questions:**

What is the purpose of \hat{K}-NMF?

---

> ### Author Response · Authors · 2023-11-13
> **Response to Reviewer oPxL**
>
> **[Related work and comparisons]** &emsp; We thank the reviewer for the kind summary. However, we want to point out the fundamental difference between our work and Poisson Factorization, Bayesian MF, and other methods mentioned by the reviewer.
>
> Poisson Factorization, as used for example in recommendation systems (e.g., [1]), introduces k latent features (preferences) of the user and parallel k features (attributes) of the item. The product of those (followed by Poisson noise) determines the observation.
>
> As a plate model, this can be represented the same as the NMF model (Figure 1C top), where W and H are latent (sampled from priors). Importantly, the items’ attributes H can vary between items but cannot vary among users – they are assumed to be shared between all users. In contrast, in our model, the component matrix H has a different value per-user i (sample i in our setting), H[i]. In terms of the plate model shown in Figure 1C (bottom), the red plate over samples includes the variable H which is indexed by both i (samples) and j (genes).
>
> To the best of our knowledge, the various Poisson Factorization, Probabilistic MF, Bayesian MF, and Hierarchical Poisson Factorization models all share the same inherent structure, where the component matrix is shared between samples [1,2,3,4]. Therefore, they cannot capture variability in sources signal between samples. Thus, adding additional comparisons will not touch the main point which is the usefulness of capturing variability in sources signal between samples and the feasibility of estimating it from mixed samples.
>
> That said, following the comments of this reviewer and the others, we will revise the related work section to further highlight this difference. Please let us know if you have more questions concerning this issue.
>
> **[Different datasets]** &emsp; Our work focuses on real-world applications. Specifically, we are mostly interested in the computational biology domain that offers a wide range of possible applications. While the suggested applications are of interest, we decided to analyze problems in this domain only.
>
> **[Robustness to noise]** &emsp; We agree with the reviewer that robustness is an important property and we performed various evaluations. We decided to focus our presentation in the paper on the one we believed is most crucial – robustness to increased level of source variability. We will add to the appendix results regarding levels of observation noise and number of samples.
>
> **[Reproducibility Package]** &emsp; We agree with the reviewer. We missed the instructions for the Reproducibility Package in the instructions for authors. We were planning to make the code and data available with publication of the paper. We will add a Reproducibility Statement and Package.
>
> **[Title]** &emsp; We oscillated between having a detailed title and a shorter one. We can shift back to “Non-negative probabilistic factorization with source variability”.
>
>  &emsp;
>
> Regarding your question:
>
> **[$\tilde{K}$-NMF]** &emsp; As mentioned in Section 4.2, the number of parameters in VarNMF is larger than in NMF. To show that the improvement in performances in VarNMF is not due to this increase, we also test it against NMF with an increased number of sources - $\tilde{K}$, where $\tilde{K}$ is the minimal number of sources to compensate for the difference in number of parameters.
>
>
> &emsp;
>
> References:
>
> [1] Gopalan, P., Hofman, J. M., & Blei, D. M. (2013). Scalable recommendation with poisson factorization. arXiv preprint arXiv:1311.1704.
>
> [2] Mnih, A., & Salakhutdinov, R. R. (2007). Probabilistic matrix factorization. Advances in neural information processing systems, 20.
>
> [3] Salakhutdinov, R., & Mnih, A. (2008, July). Bayesian probabilistic matrix factorization using Markov chain Monte Carlo. In Proceedings of the 25th international conference on Machine learning (pp. 880-887).
>
> [4] Gopalan, P., Hofman, J. M., & Blei, D. M. (2015, July). Scalable Recommendation with Hierarchical Poisson Factorization. In UAI (pp. 326-335).

---

### Official Review · Reviewer_aHM6 · 2023-11-01

**Soundness:** 3 good
**Presentation:** 2 fair
**Contribution:** 2 fair
**Rating:** 5
**Confidence:** 3

**Summary:**

The manuscript proposes a probabilistic version of nonnegative matrix factorization (NMF) where the latent factor factors (H or the basis matrix) are assumed to be sampled per data point. This modeling leads to the implementation (by expectation and maximization) that each data point can have its own H-matrix as a parameter. The proposed method is applied to a cfChIP-seq dataset to extract inter-cancer variability.

**Strengths:**

By relaxing the assumption of standard NMF that the constant components (H matrix) are shared between data points, the proposed factorization model can capture the variations in the signals between data points.

**Weaknesses:**

1. I am concerned about the high computational complexity of the proposed model (the number of parameters increases with the number of data points).

2. I think that the presentation could be further improved.
1) The symbols are used inconsistently. For example, the symbol R is used to represent a function and a variable; it seems that there are two indexing systems used for matrices. V[i]_j and H_{k,j} in eq. (2), but H[i] also represents a matrix in eq. (3).
2) From the introduction and the model descriptions, it seems that the proposed method aims at solving source separation or deconvolution. For example, "the objective of NMF is to decompose the mixed single into K sources...  " in Section 2. If this is what the authors intended to claim, the authors should provide rigorous analysis or experiments to prove whether the proposed method can recover the sources from the mixed signals. If not, I think the current presentation is confusing and should be modified.

3. The manuscript does not include extensive or rigorous experiments with real data. In section 4.3, it is stated that the experiment was done once with the randomly split dataset.

**Questions:**

Regarding the training/testing experiments in sections 4.2 and 4.3. Each data point has its H[i] matrix. How can we define H[*] for a test point *?

Have you compared the proposed NMF method with other Bayesian NMF models on the dataset used in the experiments?

---

> ### Author Response · Authors · 2023-11-13
> **Response to Reviewer aHM6**
>
> We thank the reviewer for these comments.
>
> **[High computational complexity]** &emsp; We believe that this comment, as well as the first question, is due to a confusion between the model parameters and the latent variables in the model. The individual H[i] of each sample i is a random variable that we integrate over. The parameters of the model $\theta_H$ define the prior distribution over H, from which H[i] are assumed to be sampled i.i.d. These distinctions are highlighted in the plate model of Figure 1C. Specifically, the number of parameters in VarNMF is NK+2MK vs NK+MK in NMF.
>
> Does this answer your concern?
>
> **[Notation]** &emsp; We are sorry that the notation is not clear. We are adding a notation section to provide a clear reference for the notation we use. We hope that this will alleviate the confusion.
>
> **[Deconvolution & source separation]** &emsp; The reviewer mentions the evaluation of the ability to recover the unknown H from data. This is precisely the goal of the synthetic experiments where we show the difference between the true H to the reconstructed prior H under a range of conditions (Figure 2B).
>
> **[Experiments in real data]** &emsp; Our evaluation on real data aims to provide understanding of the implication of the learned models beyond performance (that was evaluated on synthetic data). Thus, our focus was on the biological relevance of the learned components. We can of course provide results for multiple train/test splits.
>
> &emsp;
>
> Regarding the questions:
>
> **[Test component matrix]** &emsp; For a training sample i, given the learned weights W[i] and prior distribution over H, we estimate H[i] using the mean of the per-sample posterior distribution from Eq.18 (see illustration in Figure 2C). For a test sample \*, as described in Appendix A.7, the weights W[\*] are estimated using the EM procedure while keeping the prior distributions over H constant (similar to the deconvolution problem). Given the new weights W[\*], estimating H[\*] is done in the same way as for a training sample. Please let us know if this is still not clear.
>
> **[Comparison to Bayesian NMF]** &emsp; While we thank the reviewer for the suggestion of additional baselines, we note that similar to NMF, using Bayesian NMF models will result in an estimation of a component matrix H that is shared between all samples. Although the extracted H has the potential to result in a more accurate estimation for the mean signal of the sources, it cannot capture the variability in sources signal between samples. Thus, adding additional comparisons will not touch the main point which is the usefulness of capturing variability in sources signal between samples and the feasibility of estimating it from mixed samples.

---

### Author Response · Authors · 2023-11-23
**Revision summary**

We uploaded revised manuscript with the following changes:
* Fixed title.
* Fixed typos.
* Added to the Appendix results on synthetic data with varying observation noise and number of samples.
* Added formulation of Bayesian NMF to Figure 1.
* Results on real data show 5 random splits of train/test data.
* Notation section.
* Added Reproducability Statement.
* Minor clarifications in the description of the data.

We also uploaded a reproducibility package with code to generate the results.

---

### Author Response · Authors · 2023-11-23
**A more general comment about the reviewer process**

Dear Reviewers and Committee Members,

This is the senior author with some high level comments about the discussion here. I believe that anonymity restrictions allow me to say that in my past I participated as committee member and section/program chair in several AI/ML conferences. I apologise if this came out a bit long.

As one who did not publish in ICLR before, I did not have a clear idea of what to expect from the reviews and this discussion. I like the iterative discussions and believe they are an opportunity to have a somewhat more balanced exchange between the authors and the reviewers.

A good review process is one that serves two overlapping functions. From the conference perspective it should identify the most relevant/excellent/solid/important manuscript for participation in the meeting. From the author’s perspective it is a chance to get unfiltered but hopefully constructive critique that will allow us to improve our science. In my view these two functions are tied together, in that a good constructive review is one that shows the program chairs what are the merits and shortcomings of the paper and allow them to balance these in the bigger view of other submissions. Except for extreme cases, a terse review that does not provide information is also one that is not useful for the purposes of decision making.

Some of the critique we received was relevant and important: extent of validations, typos, clarity of notations, and even the title (We shortened the title due space squeeze as  the huge font caused the original title to take the space of a whole paragraph). In few of these cases we already did what the reviewer asked for and the comment was essentially about the choices we made when deciding what to include in the paper and what is “too much”.

Other critique was based on mis-understanding of some of the points in manuscript. My view is that these reflect a failure on our part in the presentation, and as such it is also useful. Even if the reader skimmed the manuscript, the key ideas should pop out.

Finally, there are critiques that I find unusefull. Comparison to relevant literature is important, but especially in a conference format should focus on the most crucial aspects. Had we tried to improve on a task that has been addressed in the literature before, we definitely need to discuss and empirically compare to relevant methods. This is not the case here. We found a deficiency in the ability to extract useful insights from NMF-based investigation of complex real-life data. We explained the basis of that deficiency, showed an approach to address it, and how it relates to actual properties of the real-life data. In such a situation the right straw-man is the plainest, most understood method (“plain” NMF) and not the latest and greatest variants if these variants do not deal with the key issues we are trying to solve. Had the graphs included five more lines with different Bayesian NMF it would still be the case the final estimate would be a point source (MAP or integral over posterior), and would not allow us to understand how sources change between samples (e.g., before/after cancer treatment). For this reason I find the exchange about related work unconstructive and in fact mainly a sign that the reviewers are focused more on finding reasons to reject than understanding merits and drawbacks of a paper.

An additional note is on the respect between scientists. Writing an anonymous review is often a trap for writing dismissive and disrespectful comments. As a general rule, my recommendation is always to write the review as though it was signed but not to hold back on factual critique. I find the comment “If the authors spend a little more time on the work [TF12], they…” to be disrespectful. We read the paper, and while we felt that reviewer did not bother to read our manuscript when writing gross mischaracterization of some of the formula, we kept in mind the possibility that we were not clear and answered respectfully and with a detailed discussion (which I am not convinced the reviewer read before answering).

Sincerely

Anonymous author

---

### Meta-Review · Area_Chair_6iaG · 2023-12-07

**Metareview:**

This paper introduces VarNMF, a probabilistic extension of Non-negative Matrix Factorization (NMF) that captures an additional type of variation in mixed samples: the variation in source contributions. It models sources as distributions and uses Expectation Maximization to learn this variation without direct source observations. Applied to genomic data from liquid biopsies, VarNMF extracts cancer-associated source distributions reflecting inter-cancer variability from mixed samples without prior knowledge, providing a novel framework for learning source distributions from mixed data.

The paper underwent review by four experts, and it faced several concerns. One major issue raised by all reviewers was the lack of an adequate discussion on related work. This omission was considered a major weakness, as it hindered the paper's ability to contextualize itself within the existing literature, particularly in the field of probabilistic/Bayesian NMF. One reviewer specifically pointed out a close connection between the proposed work and that of Tan and Févotte (2012) and requested a more thorough comparison, which the reviewer found lacking in the authors' response. Regrettably, I concur with the reviewer's assessment, and I believe that the paper requires a substantial revision to address these concerns.

**Justification For Why Not Higher Score:**

Several reviewers highlighted the lack of comparison to other related work.

**Justification For Why Not Lower Score:**

N/A

---

### Decision · Program_Chairs · 2024-01-16

Reject